# A conserved strategy for inducing appendage regeneration in moon jellyfish, *Drosophila*, and mice

**Michael J Abrams**[1†‡§#¶]**, Fayth Hui Tan**[1†]**, Yutian Li**[1†]**, Ty Basinger**[1]**, Martin L Heithe**[1]**, Anish Sarma**[1]**, Iris T Lee**[1]**, Zevin J Condiotte**[1]**, Misha Raffiee**[1‡§#¶]**, John O Dabiri**[2]**, David A Gold**[1‡§#¶]**, Lea Goentoro**[1]*****

[1]Division of Biology and Biological Engineering, California Institute of Technology, Pasadena, United States; [2]Graduate Aerospace Laboratories and Mechanical Engineering, California Institute of Technology, Pasadena, United States

**\*For correspondence:**
goentoro@caltech.edu

[†]These authors contributed equally to this work

**Present address:** [‡]Department of Molecular and Cell Biology, University of California, Berkeley, Berkeley, United States; [§]Department of Bioengineering, Stanford University, Stanford, United States; [#]Department of Biology and Allied Health Sciences, Bloomsburg University, Pennsylvania, United States; [¶]Department of Earth and Planetary Sciences, University of California, California, California, United States

**Abstract** Can limb regeneration be induced? Few have pursued this question, and an evolutionarily conserved strategy has yet to emerge. This study reports a strategy for inducing regenerative response in appendages, which works across three species that span the animal phylogeny. In Cnidaria, the frequency of appendage regeneration in the moon jellyfish *Aurelia* was increased by feeding with the amino acid L-leucine and the growth hormone insulin. In insects, the same strategy induced tibia regeneration in adult *Drosophila*. Finally, in mammals, L-leucine and sucrose administration induced digit regeneration in adult mice, including dramatically from mid-phalangeal amputation. The conserved effect of L-leucine and insulin/sugar suggests a key role for energetic parameters in regeneration induction. The simplicity by which nutrient supplementation can induce appendage regeneration provides a testable hypothesis across animals.

## Editor's evaluation

This paper shows that simple nutritional interventions such as L-leucine, insulin or sucrose can trigger appendage regeneration in three species that do not regenerate appendage in normal conditions, the Aurelia jellyfish, *Drosophila* flies and mice. The results are stunning and provide novel model systems to induce appendage regeneration in animals and to study the mechanisms underlying regeneration.

## Introduction

In contrast to humans' poor ability to regenerate, the animal world is filled with seemingly Homeric tales: a creature that regrows when halved or a whole animal growing from a small body piece. Two views have historically prevailed as to why some animals regenerate better than others (*Goss, 1992*). Some biologists, including Charles Darwin and August Weismann, hold that regeneration is an adaptive property of a specific organ (*Polezhaev, 1972*). For instance, some lobsters may evolve the ability to regenerate claws because they often lose them in fights and food foraging. Other biologists, including Thomas Morgan, hold that regeneration is not an evolved trait of a particular organ, but inherent in all organisms (*Morgan, 1901*). Regeneration evolving for a particular organ versus regeneration being organismally inherent is an important distinction, as the latter suggests that the lack of regeneration is not due to the trait never having evolved, but rather due to inactivation—and may therefore be induced. In support of Morgan's view, studies in past decades have converged on one striking insight: many animal phyla have at least one or more species that regenerate body parts (*Sánchez Alvarado,*

**eLife digest** The ability of animals to replace damaged or lost tissue (or 'regenerate') is a sliding scale, with some animals able to regenerate whole limbs, while others can only scar. But why some animals can regenerate while others have more limited capabilities has puzzled the scientific community for many years. The likes of Charles Darwin and August Weismann suggested regeneration only evolves in a particular organ. In contrast, Thomas Morgan suggested that all animals are equipped with the tools to regenerate but differ in whether they are able to activate these processes. If the latter were true, it could be possible to 'switch on' regeneration.

Animals that keep growing throughout their life and do not regulate their body temperatures are more likely to be able to regenerate. But what do growth and temperature regulation have in common? Both are highly energy-intensive, with temperature regulation potentially diverting energy from other processes. A question therefore presents itself: could limb regeneration be switched on by supplying animals with more energy, either in the form of nutrients like sugars or amino acids, or by giving them growth hormones such as insulin?

Abrams, Tan, Li et al. tested this hypothesis by amputating the limbs of jellyfish, flies and mice, and then supplementing their diet with sucrose (a sugar), leucine (an amino acid) and/or insulin for eight weeks while they healed. Typically, jellyfish rearrange their remaining arms when one is lost, while fruit flies are not known to regenerate limbs. House mice are usually only able to regenerate the very tip of an amputated digit. But in Abrams, Tan, Li et al.'s experiments, leucine and insulin supplements stimulated limb regeneration in jellyfish and adult fruit flies, and leucine and sucrose supplements allowed mice to regenerate digits from below the second knuckle. Although regeneration was not observed in all animals, these results demonstrate that regeneration can be induced, and that it can be done relatively easily, by feeding animals extra sugar and amino acids.

These findings highlight increasing the energy supplies of different animals by manipulating their diets while they are healing from an amputated limb can aid in regeneration. This could in the future pave the way for new therapeutic approaches to tissue and organ regeneration.

*2000*; *Bely and Nyberg, 2010*). Further, even in poorly regenerative lineages, many embryonic and larval stages can regenerate. In regenerating animals, conserved molecular events (e.g., *Cary et al., 2019*, *Kawakami et al., 2006*) and regeneration-responsive enhancers (*Wang et al., 2020*) were identified. Although the hypothesis of convergent evolution cannot be fully excluded (e.g., *Lai and Aboobaker, 2018*), these findings begin to build the case that the ability to regenerate may be ancestral (*Sánchez Alvarado, 2000*; *Bely and Nyberg, 2010*). Regeneration being possibly ancestral begs the question: is there a conserved mechanism to activate regenerative state?

This study explored how, and whether, limbs can be made to regenerate in animals that do not normally show limb regeneration. In adult frogs, studies from the early 20th century and few recent ones have induced various degrees of outgrowth in the limb using strategies including repeated trauma, electrical stimulation, local progesterone delivery, progenitor cell implantation, and Wnt activation (*Carlson, 2007*; *Lin et al., 2013*; *Kawakami et al., 2006*; *Herrera-Rincon et al., 2018*). Wnt activation restored limb development in chick embryos (*Kawakami et al., 2006*), but there are no reports of postnatal regeneration induction. In salamanders, a wound site that normally just heals can be induced to grow a limb by supplying nerve connection and skin graft from the contralateral limb (*Endo et al., 2004*), or by delivery of Fgf2, 8, and Bmp2 to the wound site followed by retinoic acid (*Vieira et al., 2019*). In neonatal and adult mouse digits, a model for exploring limb regeneration in mammals, bone outgrowth, or joint-like structure can be induced via local implantation of Bmp2 (bone) or Bmp9 (joint; *Yu et al., 2019*). Thus far, different strategies gain tractions in different species, and a common denominator appears elusive.

However, across animal phylogeny, some physiological features show interesting correlation with regenerative ability (*Hariharan et al., 2016*; *Vivien et al., 2016*; *Sousounis et al., 2014*). First, regeneration especially in vertebrates tends to decrease with age, with juveniles and larvae more likely to regenerate than adults. For instance, the mammalian heart rapidly loses the ability to regenerate after birth and anurans cease to regenerate limbs upon metamorphosis. Second, animals that continue to grow throughout life tend to also regenerate. For instance, most annelids continue adding body

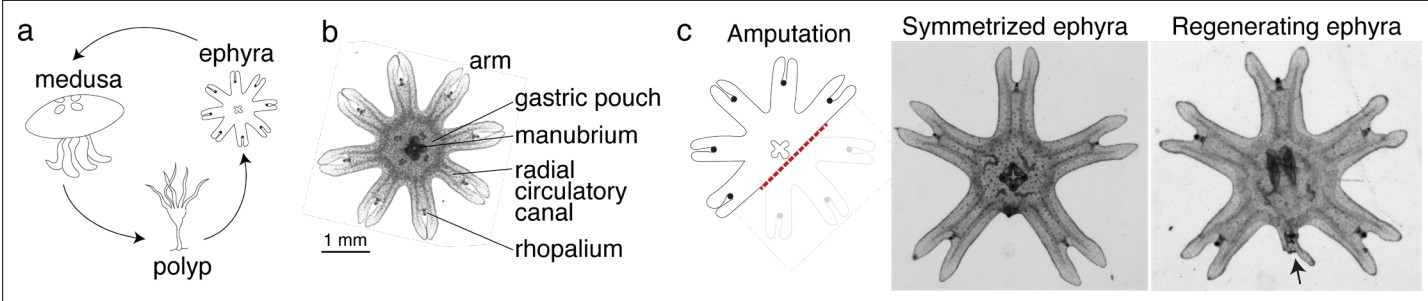

**Figure 1.** Aurelia as a system to identify factors that promote appendage regeneration. (**a**) The moon jellyfish *Aurelia aurita* has a dimorphic life cycle, existing as sessile polyps or free-swimming medusae and ephyrae. Ephyra is the juvenile stage of medusa, a robust stage that can withstand months of starvation. In lab conditions, ephyrae mature into medusae, growing bell tissue, and reproductive organs, in 1–2 months. (**b**) Ephyrae have eight arms, which are swimming appendages that contract synchronously to generate axisymmetric fluid flow, which facilitates propulsion and filter feeding. The eight arms are symmetrically positioned around the stomach and the feeding organ manubrium. Extending into each arm is radial muscle (shown in *Figure 2*) and a circulatory canal that transports nutrients. At the end of each arm is the light- and gravity-sensing organ rhopalium. (**c**) In response to injury, the majority of ephyrae rapidly reorganize existing body parts and regain radial symmetry. However, performing the experiment in the natural habitat, a few ephyrae (2 out of 18) regenerated a small arm (arrow).

segments and regenerate well, a striking exception of which is leeches that make exactly 32 segments and one of the few annelids that do not regenerate body segments (*Rouse, 1998*). Consistent with the notion of regeneration as ancestral, indeterminate growth is thought of as the ancestral state (*Hariharan et al., 2016*). Finally, a broad correlate of regenerative ability across animal phylogeny is thermal regulation. Poikilotherms, which include most invertebrates, fish, reptiles, and amphibians, tend to have greater regenerative abilities than homeotherms—birds and mammals are animal lineages with poorest regeneration. These physiological correlates, taken together, are united by the notion of energy expenditure. The transition from juvenile to adult is a period of intense energy usage, continued growth is generally underlined by sustained anabolic processes, and regulating body temperature is energetically expensive compared to allowing for fluctuation. Regeneration itself entails activation of anabolic processes to rebuild lost tissues (*Hirose et al., 2014*; *Naviaux et al., 2009*; *Malandraki-Miller et al., 2018*; *Takayama et al., 2018*). These physiological correlates thus raise the notion of a key role of energetics in the evolution of regeneration in animals. Specifically, we wondered whether energy inputs can promote regenerative state. In this study, we demonstrate that nutrient supplementation can induce regenerative response in appendage and limb across three vastly divergent species.

## Results

### Leucine and insulin promote appendage regeneration in the moon jelly *Aurelia*

We reasoned that if there was an ancestral mechanism to promote regeneration, it would likely remain intact in early branching lineages with prevalent regeneration across the species. In Cnidaria, the ability to regenerate is established in polyps, for example, hydras and sea anemones. Some cnidarians, notably jellyfish, not only exist as sessile polyps, but also as free-swimming ephyrae and medusae (*Figure 1a*). In contrast to the polyps' ability to regenerate, regeneration in ephyrae and medusae appears more restricted in some species (*Abrams et al., 2015*; *Sinigaglia et al., 2020*; *Schmid and Alder, 1984*). We focused on the moon jellyfish *Aurelia coerulea* (formerly *A. aurita* sp. 1 strain), specifically on the ephyra, whose eight arms facilitate morphological tracking (*Figure 1b*). About 3 mm in diameter, *Aurelia* ephyrae regenerate the tips of arms and the distal sensory organ rhopalium, but upon more dramatic amputations such as removing a whole arm or halving the body, rapidly reorganize existing body parts and regain radial symmetry (*Figure 1c*). Observed across four scyphozoan species, symmetrization occurs rapidly within 1–3 days and robustly across conditions (*Abrams et al., 2015*). Ephyrae that symmetrized matured into medusae, whereas ephyrae that failed to symmetrize and simply healed the wound grew abnormally.

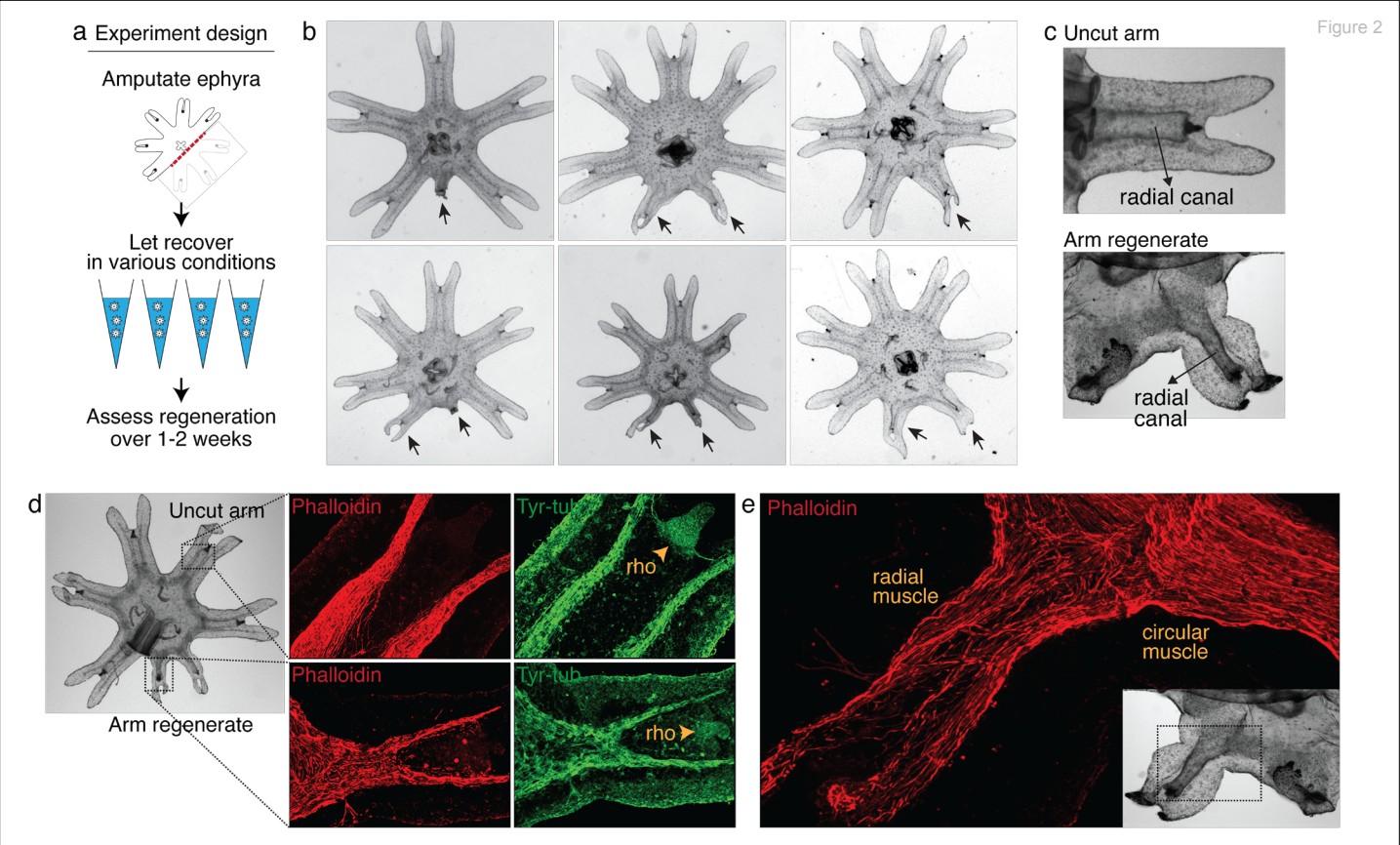

**Figure 2.** Arm regeneration in *Aurelia* ephyra can be induced using exogenous factors. (**a**) Ephyrae were amputated (red line) across the body to remove three arms, and then let recover in various conditions. *Figure 2—source data 1* tabulates the factors tested in the screen. Regeneration was assessed over 1–2 weeks until bell tissues began developing between the arms and obscured scoring. The ephyrae shown are from high-nutrient condition (see *Figure 3*). (**b**) Arm regeneration (arrows; from high food condition, see *Figure 3a*). (**c**) Radial circulatory canal in an uncut arm and is reformed in an arm regenerate. (**d**) Muscle (red), as indicated by phalloidin staining, and neuronal networks (green), as indicated by antibody against tyrosinated tubulin. The orange arrows indicate distal enrichment of tyrosinated-tubulin staining, which marks the sensory organ rhopalium (rho). Twenty ephyrae were examined and representative images are shown. (**e**) Higher magnification of the phalloidin staining shows the striated morphology of the regrown muscle in the arm regenerate (called radial muscle), which extends seamlessly from circular muscle in the body.

The online version of this article includes the following figure supplement(s) for figure 2:

**Figure supplement 1.** Bell growth limited the time window for assessing arm regeneration.

**Figure supplement 2.** Variable extent of regeneration was observed in clonal lines.

**Source data 1.** This spreadsheet details the factors screened in *Aurelia*, rationale for screening the factors, targets of the molecular modulators tested, doses or parameters tested, estimate number of ephyrae tested, and relevant references.

Intriguingly, in the course of our previous study (*Abrams et al., 2015*), we observed in a few symmetrizing ephyrae, a small bud at the amputation site. To follow this hunch, we repeated the experiment in the original habitat of our lab's polyp population, off the coast of Long Beach, CA (Materials and methods). Two weeks after amputation, most ephyrae indeed symmetrized, but in 2 out of 18 animals a small arm grew (*Figure 1c*). This observation suggests that, despite symmetrization being the more robust response to injury, an inherent ability to regenerate arm is present and can be naturally manifest. The inherent arm regeneration presents an opportunity: Can arm regeneration be reproduced in the lab, as a way to identify factors that promote regenerative state?

To answer this question, we screened various molecular and physical factors (*Figure 2a*, *Figure 2—source data 1*). Molecularly, we tested modulators of developmental signaling pathways as well as physiological pathways such as metabolism, stress response, immune, and inflammatory response. Physically, we explored environmental parameters, such as temperature, oxygen level, and water current. Amputation was performed across the central body removing three arms (*Figure 2a*).

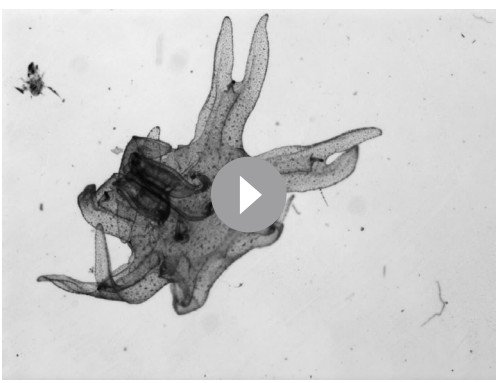

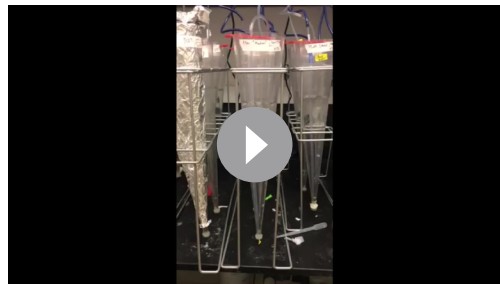

**Video 2.** Experimental setup in *Aurelia*. See *Figure 3— figure supplement 1* for more details.
https://elifesciences.org/articles/65092/figures#video2

**Video 1.** Arm regenerates pulse synchronously with existing arms. This video was taken 2 weeks after amputation. The synchronous pulsing suggests functional rebuilding of neuromuscular tissues.
https://elifesciences.org/articles/65092/figures#video1

Parameter changes were implemented or molecular modulators (e.g., peptides and small molecules) were introduced into the water immediately after amputation. Regenerative response was assessed for 1–2 weeks until the onset of bell growth, which hindered the scoring of arm regeneration (*Figure 2—figure supplement 1*).

After 3 years of screening, only three factors emerged that strongly promoted arm regeneration (*Figure 2b*). The ephyrae persistently symmetrized in the majority of conditions tested. In the few conditions where regeneration occurred, arm regenerates show multiple tissues regrown in the right locations: circulatory canals, muscle, neurons, and rhopalium (*Figure 2c–e*). The arm regenerates contracted synchronously with the original arms (*Video 1*), demonstrating a functional neuromuscular network. Thus, arm regeneration in *Aurelia* that was observed in the natural habitat can be recapitulated in the lab by administering specific exogenous factors.

The extent of arm regeneration varied, from small to almost fully sized arms (*Figure 2b*). The variation manifested even within individuals: a single ephyra could grow differently sized arms. Of the three arms removed, if regeneration occurred, generally one arm regenerated (67%), occasionally two arms (32%), and rarely three arms (1%, of the 4270 total ephyrae quantified in this study). Finally, the frequency of regeneration varied across clutches, that is, strobilation cohorts. Some variability may be due to technical factors, for example, varying feed culture conditions; however, variability persisted even with the same feed batch. We verified that the variability was not entirely due to genetic differences, as it manifested across clonal populations (*Figure 2—figure supplement 2*). Thus, there appears to be stochasticity in the occurrence of arm regeneration in *Aurelia* and the extent to which regeneration proceeds.

What are the factors that promote arm regeneration? Notably, modulation of developmental pathways often implicated in regeneration literature (e.g., Wnt, Bmp, and Tgfß) did not produce effect in the screen (*Figure 2—source data 1*)—although we do not rule out their involvement in other capacity, for example, in downstream patterning. We first identified a necessary condition: water current. The requirement for current for promoting regeneration is interesting because ephyrae can recover from injury by symmetrizing in stagnant water (*Figure 1c*). Thus, a specific physiological state is required for enabling regenerative response. Behaviorally, the presence of current promotes more swimming, while in stagnant water ephyrae tend to rest at the bottom and pulse stationarily (*Figure 3—figure supplement 1* and *Video 2* show the aquarium setup used to implement current). In this permissive condition, the first factor that promotes regeneration is the nutrient level: increasing food amount increases the frequency of arm regeneration. To measure the regeneration frequency, we scored any regenerates with lengths greater than 15% of that of an uncut arm (*Figure 3a*). This threshold was chosen to predominantly exclude non-specific growths or buds that show no morphological structures (*Figure 3b*) while including small arm regenerates that show clear morphological features, that is, lappets, radial canal, and radial muscle sometimes showing growing ends (*Figure 3b*). Given the clutch-to-clutch variability, control and treatment were always performed side by side using ephyrae from the same clutch. The effect size of a treatment was assessed by computing the change in regeneration frequency relative to the internal control. Statistical significance of a treatment was

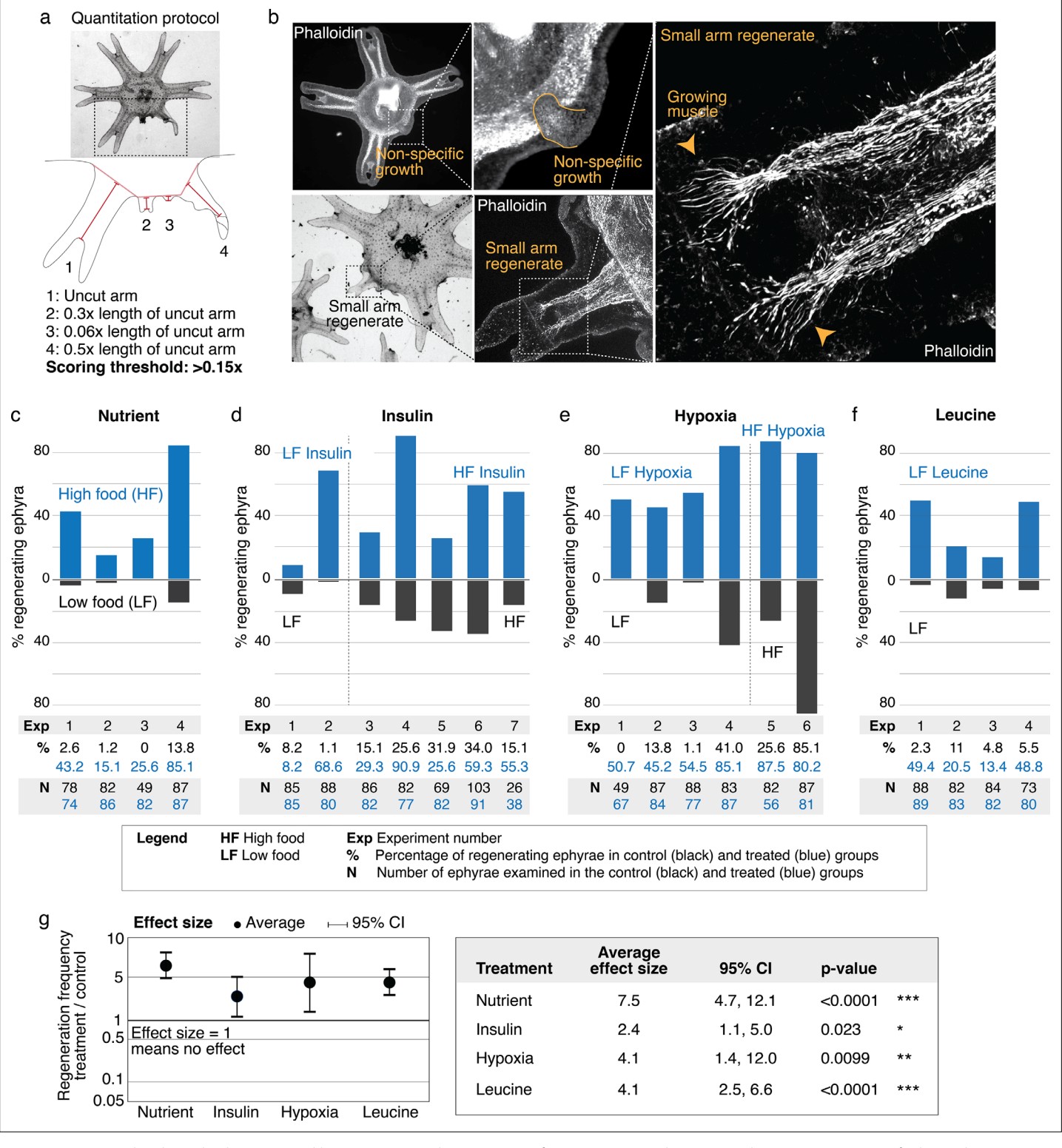

**Figure 3.** Nutrient level, insulin, hypoxia, and leucine increased regeneration frequency in *Aurelia*. (**a**) An ephyra is regenerating if it has at least one growth from the cut site with a length greater than 0.15 of the uncut arm length. The uncut arm length was determined in each ephyra by measuring three uncut arms and taking the average. Lappets, the distal paired flaps, were excluded in the length measurement because their shapes tend to vary across ephyrae. The measurements were performed in ImageJ. (**b**) The threshold 0.15 was chosen to balance excluding non-specific growths that show no morphological structures (e.g., as shown, lack of phalloidin-stained structures) and retaining rudimentary arms that show morphological structures, including radial muscle sometime with growing ends (shown, phalloidin stained). (**c–f**) In each experiment, treated (blue) and control (gray) ephyrae

*Figure 3 continued on next page*

*Figure 3 continued*

came from the same strobilation. (**c**) Regeneration frequency in lower amount of food (LF) and higher amount of food (HF). The designation 'high' and 'low' is for simplicity, and does not presume the nutrient level in the wild. If we were to speculate, the LF amount is likely closer to typical nutrient level in the wild, based on two lines of evidence. First, regeneration frequency in LF is comparable to that observed in the natural habitat experiment. Second, in many of the wild populations studied, ephyrae mature to medusae over 1–3 months (*Lucas, 2001*), comparable to the growth rate in LF (by contrast, ephyrae in HF mature to medusae over 3–4 weeks). (**d**) Regeneration frequency in 500 nM insulin. (**e**) Regeneration frequency in ASW with reduced oxygen. (**f**) Ephyrae recovering in low food, with or without 100 mM L-leucine. (**g**) The effect size of a treatment was computed from the ratio between regeneration frequency in treated and control group within an experiment, that is, the metric risk ratio RR; RR=1 means the treatment has no effect (*Borenstein et al., 2009*). The statistical significance and reproducibility of a treatment were assessed by analyzing the effect size across experiments using the meta-analysis package, metafor (*Viechtbauer, 2010*), in R with statistical coefficients based on normal distribution. See Materials and methods for details. A treatment was deemed reproducible if the 95% confidence intervals (95% CIs) of RR exclude 1. The p-value evaluates the null hypothesis that the estimate RR is 1. Reproducibility and statistical significance of each treatment were verified using another common size effect metric, odds ratio (*Figure 3—figure supplement 3*).

The online version of this article includes the following source data, source code, and figure supplement(s) for figure 3:

**Source code 1.** R codes.

**Source data 1.** This spreadsheet contains the raw data of *Figure 3* and its figure supplements.

**Figure supplement 1.** Water current is a permissive requirement for arm regeneration induction.

**Figure supplement 2.** Conservation of insulin receptor and HIFα in *Aurelia*.

**Figure supplement 3.** Statistical significance of regeneration induction in *Aurelia* assessed using odds ratio (OR).

**Figure supplement 4.** Treatments using bovine serum albumin (BSA) and argon.

**Figure supplement 5.** Regeneration phenotypes.

**Figure supplement 6.** Statistical analysis of the regeneration phenotypes in high amount of nutrients, insulin, hypoxia, and L-leucine.

**Figure supplement 7.** Ephyrae in high food, insulin, or hypoxia, and L-leucine tend to be bigger in size.

assessed by evaluating the reproducibility of its effect size across independent experiments (Materials and methods). With these measurement and statistical methodologies, we found that although the baseline regeneration frequency varied across clutches, higher food amounts reproducibly increased regeneration frequency (*Figure 3c*). The magnitude of the increase varied (*Figure 3g*, 95% confidence interval [CI] [4.7, 12.1-fold]), but the increase was reproducible (95% CI excludes 1) and statistically significant ($p < 10^{-4}$).

The second factor that promotes regeneration is insulin (*Figure 3d*). We verified that the insulin receptor is conserved in *Aurelia* (*Figure 3—figure supplement 2*). Administering insulin led to a reproducible (*Figure 3g*, 95% CI [1.1, 5.0-fold]) and statistically significant ($p < 0.05$) increase in regeneration frequency. The insulin effect was unlikely to be due to non-specific addition of proteins, since bovine serum albumin (BSA) at the same molarity showed no statistically significant effect (*Figure 3—figure supplement 4*). Finally, the third promoter of regeneration is hypoxia (*Figure 3e*). We verified that the ancient oxygen sensor HIFα is present in *Aurelia* (*Figure 3—figure supplement 2*). Hypoxia led to a reproducible (*Figure 3g*, 95% CI [1.4, 12.0-fold]) and statistically significant ($p < 0.01$) increase in regeneration frequency. To reduce oxygen, nitrogen was flown into the seawater, achieving ~50% reduction in dissolved oxygen level (Materials and methods). We verified that the effect was due to reduced oxygen rather than increased nitrogen, since reducing oxygen using argon flow similarly increased regeneration frequency (*Figure 3—figure supplement 4*). The factors can act synergistically (e.g., insulin and high nutrient level), but the effect appears to eventually saturate (e.g., hypoxia and high nutrient level).

In addition to quantifying the number of ephyrae that regenerate, we further quantified the regeneration phenotypes in each ephyra, that is, the number of arms regenerating, the length of arm regenerates, and the formation of rhopalia (*Figure 3—figure supplements 5 and 6*). Nutrient level strikingly improved all phenotypic metrics: not only more ephyrae regenerated in higher nutrients, more ephyrae regenerated multiple arms, longer arms, and arms with rhopalia. Insulin and hypoxia, interestingly, show differential phenotypes. Most strikingly, while insulin induced more ephyrae to regenerate multiple arms, hypoxia induced largely single-arm regenerates, for example, hypoxia experiments 3 and 5 in *Figure 3—figure supplement 5c*. Thus, while all factors increased the probability to regenerate, they had differential effects on the regeneration phenotypes, suggesting a

decoupling to a certain extent between the regulation of the decision to regenerate and the regulation of the subsequent morphogenesis.

Of the three factors identified in the screen, nutrient input is the broadest, and prompted us to search if a more specific nutritional component could capture the effects of full nutrients in promoting regeneration. Jellyfish are carnivorous and eat protein-rich diets of zooplanktons and other smaller jellyfish (*Graham, 2001*). Notably, all three factors induced growth: treated ephyrae are larger than control ephyrae (*Figure 3—figure supplement 7*). The growth effect is interesting because of essential amino acids that must be obtained from food, branched amino acids supplementation correlates positively with protein synthesis and growth, and in particular, L-leucine appears to recapitulate most of the anabolic effects of high amino acid diet (*Lynch and Adams, 2014*; *Stipanuk, 2008*). Motivated by the correlation between growth and increased regeneration frequency, we wondered if leucine administration could promote regeneration. Animals typically have a poor ability to metabolize leucine, such that the extracellular concentrations of leucine fluctuate with dietary consumption (*Wolfson et al., 2016*). As a consequence, dietary leucine directly influences cellular metabolism. Feeding amputated ephyrae with leucine indeed led to increased growth (*Figure 3—figure supplement 6*). Assessing arm regeneration in the leucine-supplemented ephyrae, we observed a significant increase in the regeneration frequency (*Figure 3f–g*, 95% CI [2.5, 6.6-fold], $p<10^{-4}$). Furthermore, leucine treatment phenocopies the effect of high nutrients, improving all measured phenotypic metrics: increasing multi-arm regeneration, the length of arm regenerate, and the frequency of rhopalia formation (*Figure 3—figure supplements 5 and 6*).

These experiments demonstrate that abundant nutrients, the growth factor insulin, reduced oxygen level, and the amino acid L-leucine promote appendage regeneration in *Aurelia* ephyra. The identified factors are fundamental physiological factors across animals. Might the same factors promote appendage regeneration in other animal species?

## Leucine and insulin induce regeneration in *Drosophila* limb

To pursue this question, we searched for other poorly regenerating systems, which fortunately include most laboratory models. *Drosophila*, along with beetles and butterflies, belong to the holometabolans—a vast group of insects that undergo complete metamorphosis, and that as whole, do not regenerate limbs or other appendages as adults (*Das, 2015*). Larval stages have imaginal disks, undifferentiated precursors of adult appendages such as the legs and antennae, and portions of imaginal disks have been shown to regenerate (*Worley et al., 2012*). Motivated by findings in *Aurelia*, we asked if leucine and insulin administration can induce regenerative response in the limb of adult *Drosophila*. We focused on testing leucine and insulin in this study because of considerations of specificity (i.e., nutrients are broad and composition of nutritional needs vary across species), pragmatism (i.e., administering hypoxia requires more complex setups), and in the case of *Drosophila* specifically, *Drosophila* being resistant to hypoxia (*Haddad et al., 1997*).

We amputated *Drosophila* on the hindlimb, across the fourth segment of the leg, the tibia (*Figure 4a*). After amputation, flies were housed in vials with standard food (control) or standard food supplemented with leucine and insulin, with glutamine to promote leucine uptake (*Nicklin et al., 2009*) (treated) (*Figure 4c*). Each fly was examined multiple times, twice in the first week, and then once weekly over the course of 2–4 weeks.

No regrown tibia was found in the 860 control flies examined (*Figure 4b–c*). In the treated flies, only tibia stumps were observed in the first week after amputation. But remarkably, at 7–21 days post-amputation (dpa), a few regrown tibias were observed (1.0%, N=387; *Figure 4d–e*). The regrown tibias culminate in reformed joints, articulating from which appears to be the beginning of a next segment. Control tibia stumps showed melanized clots at the tips within 1–3 dpa (*Figure 4b*), as expected from normal wound healing process (*Rämet et al., 2002*), while some treated tibias showed no clot (12.1%, N=387) and the tips stained positively with DAPI instead (14 out of 16 tibias examined; *Figure 4f*). Induction of regenerative response was observed across genetic backgrounds, in Oregon R, as discussed, and Canton S wild-type strains (*Figure 4—figure supplement 1a*). Reminiscent of *Aurelia*, not all regenerative response was patterned, few flies showed non-specific outgrowth (*Figure 4—figure supplement 1b*).

The bulk experiment enables us to assess a large number of flies and can capture dramatic regenerative response, that is, a fully regrown leg segment. But it can miss partial segment regrowth and

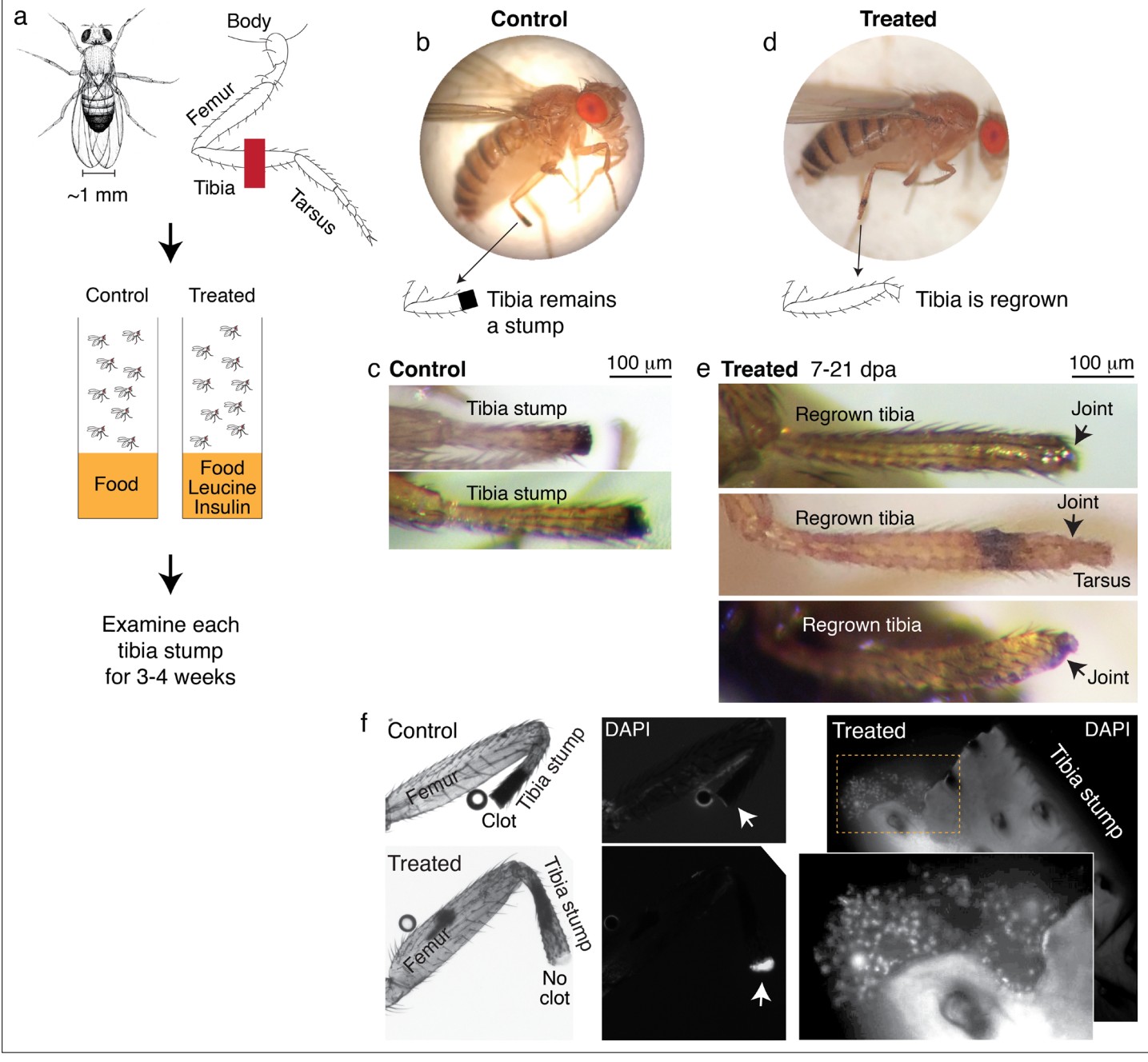

**Figure 4.** Leucine and insulin induced regeneration in *Drosophila* limb. (**a**) Adult *Drosophila* have jointed limbs, with rigid segments connected by flexible joints. Amputation was performed across the tibial segment. After amputation, flies were housed in vials containing standard lab food (control) or standard lab food added with 1.7 mM L-Leucine, 1.7 mM L-Glutamine, and 33 µg/ml insulin (treated). Doses were determined through observing the highest order of magnitude dose of amino acid that could be fed to flies over a prolonged period without shortening their lifespan. The flies were examined at ~1, 3, 7, 14, and 21 days post-amputation (dpa). (**b**) A control fly, imaged at 7 dpa. (**c**) Control tibia stumps, imaged at 1–3 dpa. (**d**) A treated fly, imaged at 7 dpa. (**e**) At 7–21 dpa, regrown tibias, which culminate in joints, were observed in the treated population. (**f**) Tibia stumps at 3–14 dpa were dissected, fixed, and mounted in Vectashield mounting medium with DAPI. Samples from 14 dpa are shown here. Insect cuticle is not dissected to restrict DAPI penetrance only to the distal tip. Clotted tips of control tibia stumps did not stain with DAPI (10 out of 10), whereas unclotted tips of treated tibia stumps were stained with DAPI (14 out of 16). Higher-resolution confocal image of an unclotted treated tibia stump at 14 dpa.

*Drosophila* drawing in (a) by Ashley Smart and used with permission.

The online version of this article includes the following figure supplement(s) for figure 4:

**Figure supplement 1.** Regenerative response induced in *Drosophila* was observed across wild-type strain.

therefore underestimate the extent of regenerative response. To track regenerative response in individual flies, we housed in each vial a small number of flies that were amputated in different limbs (*Figure 5a*). The amputated limb position, combined with sex, enabled unique identification of each fly within a vial. Each fly was imaged immediately after amputation and 1–3 additional times over the course of 2–4 weeks. The imaging frequency balanced obtaining time-lapse information with minimizing stress from repeated anesthesia.

The single-fly tracking showed pervasive regenerative response induced by nutrient supplementation. Time-lapse pictures of select control and treated flies are shown in *Figure 5b–d*. For each tibia stump, we measured the length over time (*Figure 5e*). Control tibia stumps showed near-zero percent change in length (*Figure 5d*; mean –0.3%, 95% CI [–3.8, 3.2%], N=116). By contrast, 49% of treated flies showed growth beyond the 95% CIs of the control distribution (*Figure 5f*; N=150). The rate of regrowth varied; some flies showed large growth during the first 1–2 weeks post-amputation, while others regrow more slowly over 2–3 weeks (*Figure 5c*). Interestingly, four flies (2.7%) showed shortened tibia stumps, which may indicate roles of histolysis in regenerative response. Regenerative response was not only observed from amputations across the tibia, but also from more dramatic amputations across the femur (*Figure 5d*). We verified that the control and treated length change distributions are statistically different (p<0.0001***, nonparametric Kruskal-Wallis test). No sex-based differences in response were observed (*Figure 5—figure supplement 1a*).

In some limbs, the new growth already looks almost indistinguishable from the rest of the leg. In some limbs, the new growth showed what may be intermediate morphologies. For instance, some new growths are pigmented differently and/or have no sensory bristles (*Figure 5g*, *Figure 5—figure supplement 1d, e,*) or showed white tissues protruding from the end (e.g., *Figure 5—figure supplement 1c*), which was remodeled over time (e.g., *Figure 5—figure supplement 1d*). Finally, reproducing what was observed in the bulk experiments, five flies showed reformed tibia segments: showing a tapering end (N=150; *Figure 5h*), what appeared to be a joint (*Figure 5i*), and the beginning of the next segment (*Figure 5j*).

A scanning electron micrograph of a regrown hindlimb tibia (the top tibia in *Figure 4e*, taken 1 week later) morphologically confirms the regenerated joint as a tibial/tarsal joint. The joint-like structure shows the expected bilateral symmetry of a tibial/tarsal joint (as opposed to, e.g., the radially symmetrical tarsal/tarsal joints; *Mirth and Akam, 2002*) with rounded projections at the posterior and anterior end (arrows in *Figure 5k*). These projections, called condyles, function as points of articulation between opposing leg segments. Indeed, articulating from the regrown condyles appears to be further growth of the next tarsal segment. Finally, a unique feature of the tibial/tarsal joint of the hindlimb (but not of fore or midlimb) is an additional ventral projection between the side condyles (*Mirth and Akam, 2002*), which serves to restrain bending of the leg upward. The ventral projection is indeed present in the regenerated joint (arrow in *Figure 5k*). Altogether, these data demonstrate for the first time that patterned regenerative response can be induced from adult *Drosophila* limbs.

## Leucine and sucrose induce regeneration in mouse digit

The ability of leucine and insulin to induce regenerative response in *Drosophila* limb and *Aurelia* appendage motivated testing in vertebrates. One sign that limb regeneration may be feasible in humans is that fingertips regenerate (*Illingworth, 1974*). The mammalian model for studying limb regeneration is the house mouse, *Mus musculus*, which like humans, regenerates digit tips. Although proximal regions of digits do not regenerate, increasing evidence suggests that they have inherent regenerative capacity. In adult mice, implanting developmental signals in amputated digits led to specific tissue induction, that is, bone growth with Bmp4 or joint-like structure with Bmp9 (*Yu et al., 2019*). In neonates, reactivation of the embryonic gene *lin28* led to distal phalange regrowth (*Shyh-Chang et al., 2013*). Thus, while patterned phalange regeneration can be induced in newborns, induction in adults so far involves a fine-tuned stimulation, for example, to elongate bone and then make joint, Bmp4 was first administered followed by Bmp9 in a timed manner. Motivated by the findings in *Aurelia* and *Drosophila*, we tested if leucine and insulin administration could induce a self-organized regeneration in adult mice.

We performed amputation on the hindpaw (*Figure 6a*), on digits 2 and 4, leaving the middle digit three as an internal control (*Figure 6b*). To perform non-regenerating amputation, a clear morphological marker is the nail, which is associated with the distal phalange (P3). Amputation that

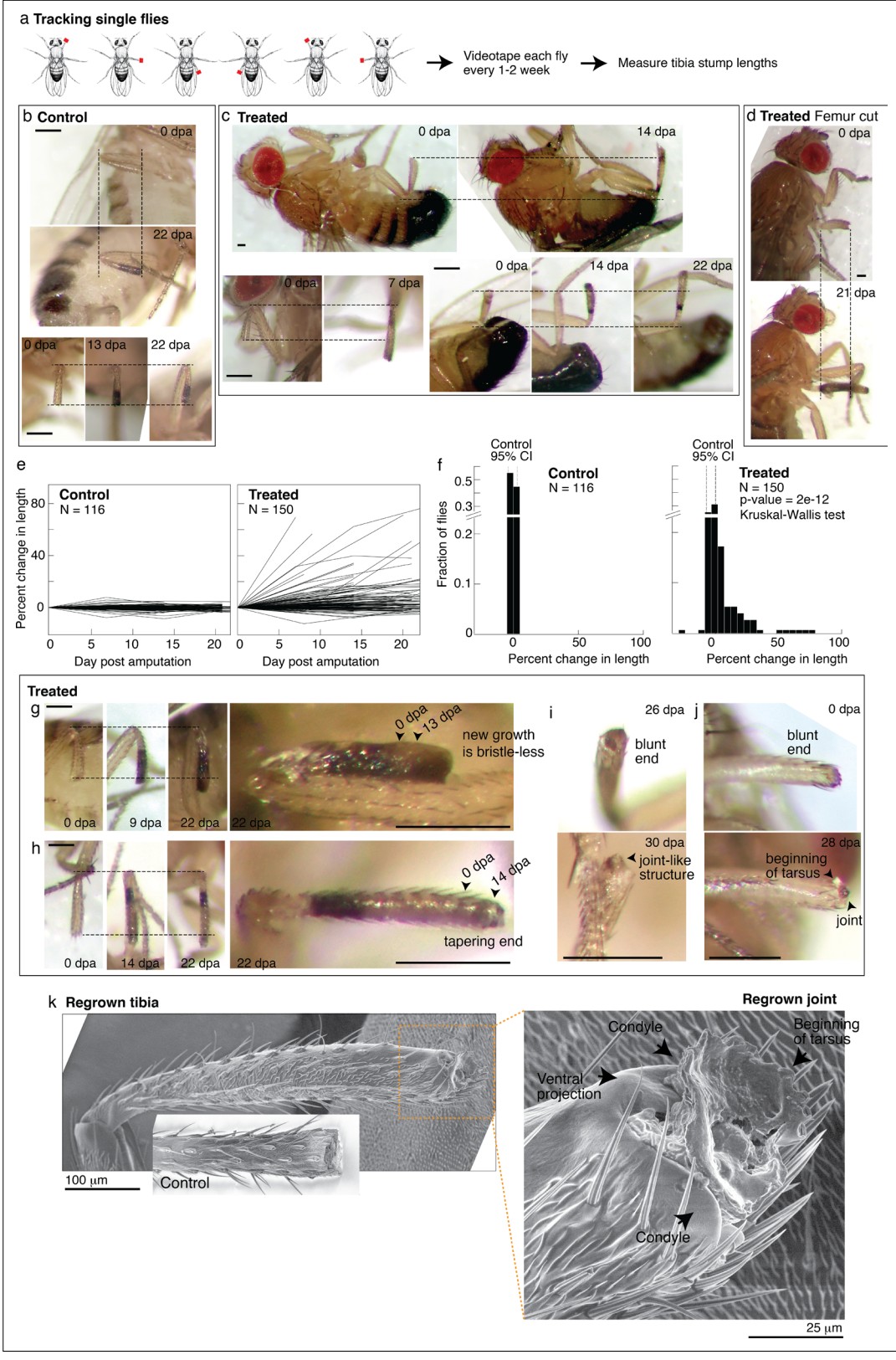

**Figure 5.** Tracking regenerative response in individual flies. (**a**) To track the regenerative response in individual flies, flies were amputated in different limbs (fore, mid, and hindlimb) and housed in small groups such that in any given vial, each fly was uniquely identifiable by sex and amputated limb. Each fly was imaged immediately after amputation and 1–3 additional times over the course of 2–4 weeks. (**b–d**) Representative time lapse from control

*Figure 5 continued on next page*

*Figure 5 continued*

(**b**) and treated flies (**c–d**). dpa: days post-amputation. For each fly, time-lapse images are shown at the same magnification. Scale bars: 250 µm. (**e**) Change in tibia stump length over time. (**f**) Distribution of change in tibia stump length in control and treated flies. Percent change in length is the difference between the length at the final time point and 0 dpa, relative to the length at 0 dpa. CI: confidence interval. Statistical difference between control and treated distributions was evaluated using the nonparametric Kruskal-Wallis test. The p-value tests the null hypothesis that the data are drawn from the same distribution. We confirmed that the probability of observing growth in each condition was not biased by differences in the initial length of tibia stump (***Figure 5—figure supplement 1b***). (**g–i**) For each fly, time-lapse images are shown at the same magnification. Scale bars: 250 µm. (**g**) In this treated tibia stump, the new growth showed different pigmentation and no sensory bristles. (**h**) In this treated tibia stump, the segment was reformed as suggested by the tapering end. (**i**) In this treated tibia stump, the segment was reformed and a joint-like structure grew. (**j**) In this treated tibia stump, amputation was performed just before the joint. After 4 weeks, a new joint-like structure appeared, from which tissues of the next segment (tarsus) started growing. (**k**) Fly with a regrown tibia at 21 dpa was mounted onto an environmental SEM with a copper stub. Inset shows a clotted tibia stump from a control fly, with the discoloration at the end corresponding to the clot. Magnification of the regenerated joint: the arrows denote the two condyles and additional ventral projection.

*Drosophila* drawing in (a) is by Ashley Smart and used with permission.

The online version of this article includes the following source data, source code, and figure supplement(s) for figure 5:

**Source code 1.** Drosophila code.

**Source data 1.** This spreadsheet contains the raw data of ***Figure 5*** and its figure supplements.

**Figure supplement 1.** Tracking regenerative response in single flies.

removes <30% of P3 length, that cuts within the nail, readily regenerates, whereas amputation that removes >60% of P3 length, corresponding to removing almost the entire visible nail, does not regenerate (***Figure 6c***; ***Chamberlain et al., 2017***; ***Lehoczky et al., 2011***). We therefore performed amputations entirely proximal to the visible nail—giving, within the precision of our amputation, a range of cut across somewhere between the proximal P3 and the distal middle phalange (P2) (***Figure 6d***)—a range that is well below the regenerating tip region. Note additional morphological markers that lie within the non-regenerating region: the os hole ('o' in ***Figure 6c***), where vasculatures and nerves enter P3, the bone marrow cavity ('m' in ***Figure 6c***), and the sesamoid bone ('s' in ***Figure 6c***) adjacent to P2.

The digit portion removed was immediately fixed to determine the precise plane of amputation. The amputated mice were either provided with water as usual (control) or water supplemented with leucine and sucrose (treated) (***Figure 6e***). Both groups were monitored for 7–8 weeks. Sucrose was used because insulin is proteolytically digested in the mammalian gut. The sucrose doses used are lower or the administration duration is shorter than those shown to induce insulin resistance (***Cao et al., 2007***; ***Togo et al., 2019***). We verified that control and treated mice had comparable initial weights (35.1±0.6 vs. 34.1±1.1 g, p=0.402, Student's t-test), and that as expected from amino acid and sugar supplementation, treated mice gained more weight over the experimental duration (4.5±1.0 vs. 7.8±1.0 g, p=0.028, Student's t-test).

As expected for amputation proximal to the nail, no regeneration was observed in the control mice (N=34 digits, 17 mice). Amputated digits healed and re-epithelialized the wound as expected (***Figure 6f***). Skeletal staining shows blunt-ended digit stumps (***Figure 6i***) and in many instances, as expected, dramatic histolysis, a phenomenon where bone recedes further from the amputation plane (***Figure 6—figure supplement 1***; ***Chamberlain et al., 2017***). By contrast, 18.8% of the treated digits (N=48 digits, 24 mice) showed various extents of regenerative response (***Figure 6—figure supplement 1***). The increase in regeneration frequency due to the treatment is statistically significant (95% CI [8, 30%], p=0.0019,**, Student's t-test).

We observed, as in *Aurelia* and *Drosophila*, an unpatterned response (***Figure 6—figure supplement 1***), wherein skeletal staining reveals excessive bone mass around the digit stump, similarly to what was observed in some cases with BMP stimulation (***Yu et al., 2019***). However, we also observed patterned responses (***Figure 6—figure supplement 2***). The most dramatic regenerative response was observed in two digits (***Figure 6g–h***). In one digit, an almost complete regrowth of the distal phalange and the nail was observed (***Figure 6g***). Skeletal staining of the portion removed from this digit (***Figure 6j***) shows that

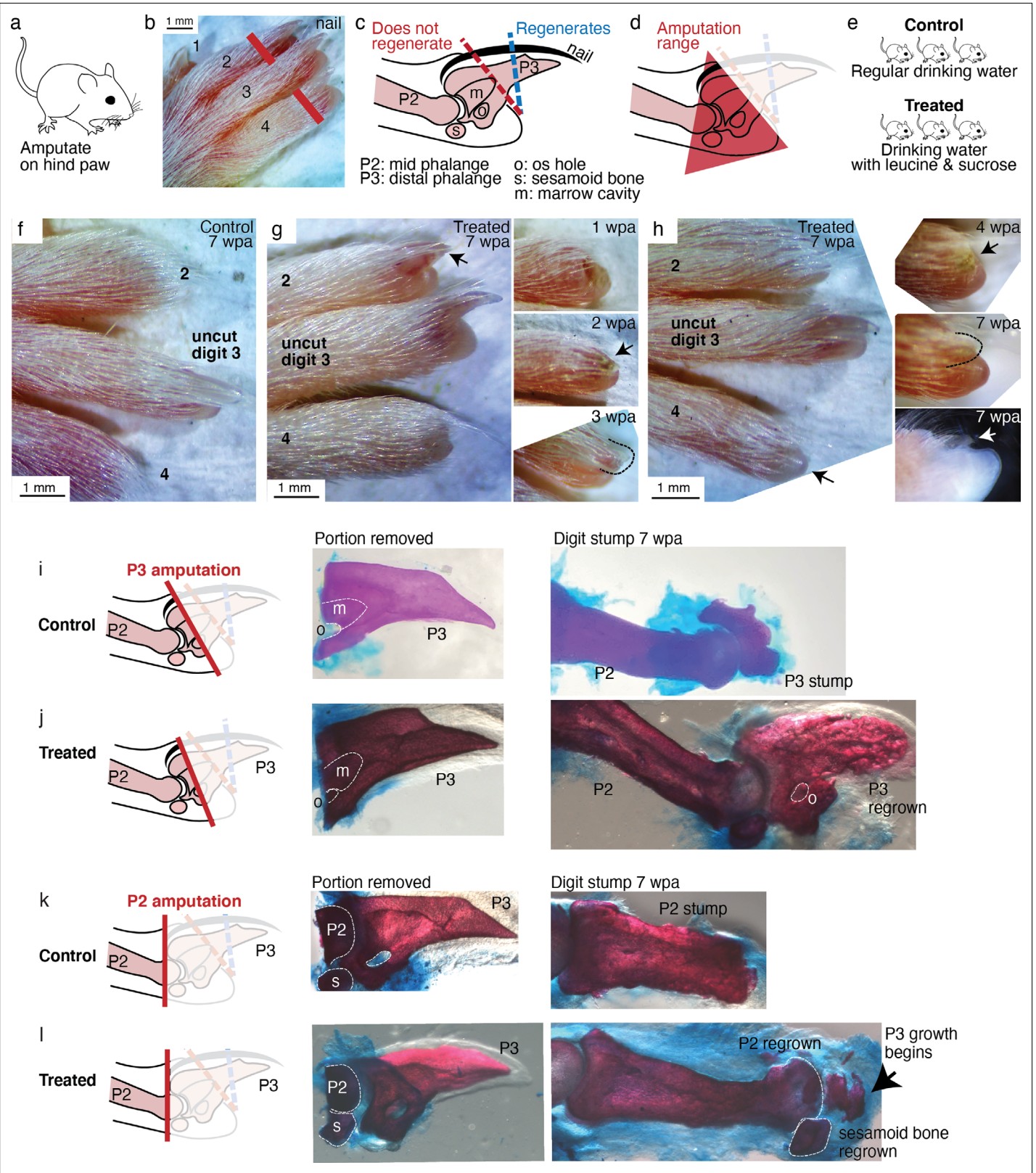

**Figure 6.** Leucine and sucrose induced regeneration in adult mouse digit. (**a–b**) Amputation was performed on hindpaws of adult (3–6 months old) mice, on digits 2 and 4, proximal to the nail. (**c**) Schematic of the distal phalange (P3) and middle phalange (P2). Amputations that remove <30% of P3 (blue line) regenerate, whereas amputations that remove >60% of P3 (red line) do not regenerate. Amputations in the intermediate region can occasionally show partial regenerative response. (**d**) Amputations in this study were performed within the red-shaded triangle. (**e**) Amputated mice were

*Figure 6 continued on next page*

*Figure 6 continued*

given regular drinking water (control) or drinking water supplemented with 1.5% L-leucine, 1.5% L-glutamine, and 4–10% w/v sucrose (2 exps with 4%, 3 exps with 10%). Drinking water, control and treated, was refreshed weekly. (**f**) A representative paw from the control group. The amputated digits 2 and 4 simply healed the wound and did not regrow the distal phalange. (**g**) In this treated mouse, digit 2 (arrow) regrew the distal phalange and nail. Insets on the right show the digit at earlier time points. At week 1, the amputation site still appeared inflamed. At week 3, the beginning of the nail appears (arrow). At week 3, a clear nail plate was observed. (**h**) In this treated mouse, digit 4 (arrow) regrew and began to show nail reformation by week 4 (top inset, see arrow), that turns into a clear nail plate by week 7 (middle inset), as can be seen more clearly from the side-view darkfield image (bottom inset). (**i–l**) Whole-mount skeletal staining. Dissected digits were stained with Alizarin red, an anionic dye that highly localizes to the bone. Left panels show illustration of the amputation plane, middle panels show skeletal staining of the portions removed, and right panels show skeletal staining of the digit stumps 7 weeks after amputation.

The online version of this article includes the following figure supplement(s) for figure 6:

**Source data 1.** This spreadsheet contains the raw data of the mouse digit phenotype in *Figure 6* and its figure supplements.

**Figure supplement 1.** Mouse digit phenotypes.

**Figure supplement 2.** Regenerative response observed in mouse digi.

it was amputated at the proximal P3 transecting the os hole. By 7 weeks, skeletal staining of the regrown digit (*Figure 6j*) shows that the P3 bone was almost completely regrown. The regrown P3 shows trabecular appearance that is similar in general structure but not identical to the original P3. Another dramatic response was observed from another digit, which began reforming the nail by 7 weeks (*Figure 6h*). Skeletal staining of the portion removed from this digit shows that it was amputated across the P2 bone, removing the entire epiphyseal cap along with the sesamoid bone (*Figure 6k*). Skeletal staining of the regenerating digit shows that the epiphyseal cap was regrown, along with its associated sesamoid bone. Moreover, articulating from the regenerated P2 appears to be the beginning of the next phalangeal bone (arrow, *Figure 6k*). To our knowledge, the regenerative response observed in these digits represents the most dramatic extent of self-organized mammalian digit regeneration reported thus far. Distal phalange regeneration in adults has not been reported, while interphalangeal joint formation from a P2 amputation has been achieved only through sequential Bmp administration (*Yu et al., 2019*) and there has been no documentation of the regrowth of the sesamoid bone.

## Discussion

In this study, amputations were performed on *Aurelia* appendage, *Drosophila* limb, and mouse digit. None of these animals are known to regenerate robustly (*Aurelia*) if at all (*Drosophila* and mouse) from these amputations. Upon administration of L-leucine and sugar/insulin, dramatic regenerative response was observed in all systems. The conserved effect of nutrient supplementation across three species that span more than 500 million years of evolutionary divergence suggests energetic parameters as ancestral regulators of regeneration activation in animals.

While we did not test the regenerative effect of hypoxia beyond *Aurelia*, it is notable that in mice hypoxia coaxes cardiomyocytes to re-enter cell cycle (*Kimura et al., 2015*) and activate HIF1α promotes healing of ear hole punch injury (*Zhang et al., 2015*). Notably in *Aurelia*, the amputation bisected through the body, and more than appendage was in fact regenerated, for example, circular muscle in the body is regrown (*Figure 2e*). Thus, nutrient supplementation may have regenerative effect in body parts beyond appendage.

The diverse physiologies of animals across phylogeny may seem difficult to reconcile with a conserved regulation of regeneration activation, especially in the view of regeneration as recapitulation of development. Growing a jellyfish appendage is different from building a fly leg or making a mouse digit. However, there is another way of looking at regeneration as a part of tissue plasticity (*Galliot and Ghila, 2010*). In this view of regeneration, upstream from tissue-specific morphogenesis is a conserved regulation of cell growth and proliferation. In support of this idea, early steps in regeneration across species and organs rely one way or another on proliferation by stem cells or differentiated cells re-entering cell cycle (*Cox et al., 2019*). We propose that in animals that poorly regenerate, high nutrient input turns on growth and anabolic states that promote tissue rebuilding upon injury.

That regenerative response can be induced seemingly blurs the boundary between regenerating versus non-regenerating animals, because the factors identified in the study are not exotic. Variations in amino acids, carbohydrates, and oxygen levels are conditions that the animals can plausibly encounter

in nature. These observations highlight two potential insights into regeneration. First, regeneration is environmentally dependent. An animal would stop at wound healing under low-energy conditions and regenerate in energy-replete conditions. In this view, for the animals examined in this study, the typical laboratory conditions may simply not be conducive to regeneration. Alternatively, the interpretation we favor, what we observed is inherent regeneration, which can be activated with broad environmental factors. We favor this interpretation because the regenerative response was unusually variable. The variability stands in stark contrast to the robust regeneration in, for example, axolotl, planaria, or hydra. Whereas wild-type processes tend to be robust, mutations produce phenotypes that are sensitive to variations in physiological parameters. Thus, just like mutant phenotypes show varying penetrance and expressivity, the variable regenerative response speaks to us as a fundamental consequence of activating a latent biological module. In this interpretation, the ordinariness of the activators suggests ancestral regeneration as part of a response to broad environmental stimuli. It would be interesting next to identify individual differences that contribute to varying propensities to mount regenerative response.

In particular, the conserved effects of nutrient supplementation suggest that regeneration might have originally been a part of growth response to abundant environments. No nutrient dependence has been observed in highly regenerating animal models such as planaria, hydra, and axolotl. Environment-dependent plasticity, however, is pervasive in development, physiology, behavior, and phenology (*West-Eberhard, 2003*; *Moczek et al., 2011*). We therefore conjecture that environment-dependent plasticity may have characterized the ancestral form of regeneration. In this conjecture, present regenerating lineages might have decoupled the linkage with environmental input and genetically assimilated regenerative response—because regeneration is adaptive or coupled to a strongly selected process, for example, reproduction. In parallel, non- or poorly regenerating animals might have also weakened the linkage with environmental input, but to silence the regenerative response. This predicts an ancient form of a robustly regenerative animal (like planaria, hydra, and axolotl) that tunes its regeneration frequency to nutrient abundance. Such plasticity has been reported in the basal lineage Ctenophora (*Bading et al., 2017*).

In conclusion, this study suggests that an inherent ability for appendage regeneration is retained in non-regenerating animals and can be unlocked with a conserved strategy. The treatments across species were not exactly identical, and correspondingly there might be differences in the precise molecular mechanisms—in spite of which they could be applied across species in a predictive manner. In line with our findings, the role of nutrients in promoting regeneration was reported in yet another species (in the *Xenopus tadpoles*, *Williams et al., 2021*). While the observed regenerative response is not perfect, this motivates further investigation into potentially more promoting factors or the possibility of combining broad promoting factors with species- or tissue-specific morphogenetic regulators. Reiterating Spallanzani's hope, Marcus Singer supposed half a century ago that '... every organ has the power to regrow lying latent within it, needing only the appropriate "useful dispositions" to bring it out (*Singer, 1958*).' The surprise, in hindsight, is the simplicity by which the regenerative state can be promoted with ad libitum amino acid and sugar supplementation. This simplicity demonstrates a much broader possibility of organismal regeneration, and can help accelerate progress in regeneration induction across animals.

# Materials and methods

**Key resources table**

| Reagent type (species) or resource | Designation | Source or reference | Identifiers | Additional information |
|---|---|---|---|---|
| Strain, strain background (*Aurelia aurita*) | A. aurita sp. 1 | Gift from the Cabrillo Marine Aquarium,San Pedro, CA | | Alternatively named *Aurelia coerulea* (*Scorrano et al., 2016*). |
| Strain, strain background (*Drosophila melanogaster*) | OregonR | Gift from Angela Stathopolous' lab, Caltech | | RRID:BDSC_5 |
| Strain, strain background (*D. melanogaster*) | CantonS | Gift from Kai Zinn lab, Caltech | | RRID:BDSC_64349 |
| Strain, strain background (*Mus musculus*) | CD1 | Charles River Laboratories | Strain 022 | Female, 3–6 months old RRID:IMSR_CRL:022 |
| Chemical compound, drug | L-leucine | Sigma-Aldrich | L1002 | Methyl-ester hydrochloride |

*Continued on next page*

*Continued*

| Reagent type (species) or resource | Designation | Source or reference | Identifiers | Additional information |
|---|---|---|---|---|
| Chemical compound, drug | L-leucine | Sigma-Aldrich | L8000 | |
| Chemical compound, drug | L-leucine | VWR | E811 | USP grade |
| Chemical compound, drug | L-glutamine | Sigma-Aldrich | G3126 | |
| Chemical compound, drug | L-glutamine | Sigma-Aldrich | G8540 | USP grade |
| Chemical compound, drug | Sucrose | Avantor | 8360 | AR ACS grade |
| Peptide, recombinant protein | Insulin | Sigma-Aldrich | I0908 | |
| Peptide, recombinant protein | Insulin | MP Biomedicals | 0219390080 | |
| Antibody | Anti-tyrosinated alpha tubulin (Rat monoclonal) | Sigma-Aldrich | MAB1864-I | (1:200) RRID:AB_2894901 |
| Antibody | Goat anti-mouse Alexa Fluor 488 (Goat polyclonal) | Life Technologies (Thermo Fisher Scientific) | A11029 | (1:200) RRID:AB_2894900 |
| Other | Alexa Fluor 555 phalloidin | Life Technologies (Thermo Fisher Scientific) | A12379 | Histological stains (1:20) |
| Other | Hoechst 33342 | Sigma-Aldrich | B2261 | Histological stains (1:10) |
| Other | Vectashield mounting medium with DAPI | Vector | H1200 | Histological stains (1:1) |
| Other | Alizarin red | Beantown Chemical | BT144735 | Histological stains, 0.005% |
| Other | Alcian blue | Across Organics | AC40046-0100 | Histological stains, 0.015% |

Aurelia aurita.

The experiments were performed in *Aurelia aurita* sp. 1 strain, also alternatively named *A. coerulea* based on recent molecular classification (*Scorrano et al., 2016*). Polyps were reared at 68°F, in 32 ppt artificial seawater (ASW, Instant Ocean), and fed daily with brine shrimps (*Artemia nauplii*) enriched with *Nannochloropsis* algae (both from Brine Shrimp Direct). To induce strobilation, polyps were incubated in 25 M 5-methoxy-2-methyl-indole (Sigma-Aldrich M15451) at 68°F for an hour (*Fuchs et al., 2014*). Ephyrae typically began to strobilate within a week. Strobilated ephyrae were fed daily with high amount of rotifers (*Brachionus plicatilis*, Reed Mariculture) until amputation time.

## Experiment in the original habitat

The polyp population in the study arose from parental polyps collected off the coast of Long Beach, CA (33°46′04.2″N 118°07′44.2″W, GPS: 33.7678376–118.1289559). Ephyrae were amputated in location and immediately after submersed in the ocean. For submerging the amputated ephyrae in the ocean, a two-layered aquarium was custom-built. Ephyrae were placed in plastic canisters with a 7-cm diameter hole cut in the lid and covered with a 250-μm plastic screen. The canisters were then placed in a thick plastic tank fitted with a 500-μm plastic screen on top. This design offers protection to the ephyrae against predators and strong waves, while at the same time allowing exchange of water, zooplanktons, and other particulates. Ephyrae were collected after 2 weeks.

## Regeneration experiments

All experiments were performed at 68°F. Two to three days old ephyrae were anesthetized in 400 μM menthol and amputated using a razor blade mounted on an x-acto knife handle. Amputated ephyrae were let to recover in 1 L sand settling cones (Nalgene Imhoff, *Figure 3—figure supplement 1*). In each experiment, ~90 animals were amputated for each condition (e.g., 90 animals for control and 90 animals for treated). Because of the varying baseline across strobilation batches, each experiment was repeated across 2–5 strobilation batches (biological replicates). These sample sizes were chosen to obtain a 95% confidence level on the treatment effect (statistical analysis described below). Hundreds

of experimental animals were first amputated, mixed together in a beaker, and then randomly allocated to the control or treatment groups. Regeneration was assessed at various times for 1–2 weeks after amputation, before onset of maturation to medusa. All data were included in the analysis.

### Rationale for the amputation scheme

Among the possible amputation schemes, 3-arm amputation was chosen because it could be performed fastest. Removing one arm requires carefully cutting across the base of the arm while avoiding injuring the surrounding body. Removing two arms is less hard but still requires awkward positioning of the knife. Removing four arms again takes more time because it requires cutting through the large protruding manubrium, which also affects the animal's feeding ability. The fast 3-arm amputation facilitates testing hundreds of ephyrae per experiment.

### Nutrients

Amputated ephyrae were fed daily with rotifers. The number of rotifers was estimated using a six-well plate fitted with STEMgrid (the same principle as using a hemocytometer). In this study, low food was ~10–20 rotifers/ephyra and high food was ~40 rotifers/ephyra. To replicate the study, these numbers should only be used as initial estimates, as what is 'low' or 'high' food amount may easily vary across lab cultures (e.g., rotifer culture, differences across *Aurelia* strains, etc.). Most if not all rotifers were typically consumed within an hour (determined by measuring the rotifers in the water).

### Insulin

Immediately after amputation, ephyrae were placed in ASW supplemented with 500 nM human recombinant insulin (Sigma-Aldrich I0908). Insulin was refreshed weekly. To determine the concentration used, a range of concentrations, 10 nM to 3 mM, were tested. The concentration 500 nM was chosen as it maximized regeneration frequency while avoiding solubility problems. To control that the effect of insulin was not due to non-specific additions of proteins, BSA at 500 nM was tested.

### Hypoxia

Immediately after amputation, ephyrae were placed in hypoxic ASW. To create a hypoxic environment, nitrogen or argon, instead of ambient air, was pumped into the bubbler cone, beginning from the day before the experiment and maintained throughout the duration of the experiment. The bubbler cone was sealed with parafilm to maintain the lowered oxygen level. The nitrogen/argon flow was adjusted to achieve 50% reduction in the dissolved oxygen level. Dissolved oxygen level was measured using a Clark-type electrode Unisense OX-500 microsensor. The measurement was normalized to oxygen level in control ASW bubbled normally with ambient air. Oxygen measurement was performed prior to the experiment and subsequently every 3 days.

### L-leucine

Immediately after amputation, ephyrae were placed in ASW supplemented with 100 μM L-leucine (Sigma-Aldrich L1002, the cell-permeable methyl ester hydrochloride form). L-leucine was refreshed weekly. To determine the concentration used, a range of concentrations from one to hundreds of mM was tested. The concentration of 100 mM was chosen as it maximized the regeneration frequency without non-specific, negative effects.

### Statistical analysis

To assess the statistical significance of the treatments, meta-analysis of effect size was performed (*Borenstein et al., 2009*). The effect size metrics used are determined by the form of the data set. For measurements of frequencies (e.g., regeneration frequency), the data sets are in the form of a 2×2 table of dichotomous variables.

|  | # ephyrae that regenerate | # ephyrae that do not regenerate |
|---|---|---|
| Control | a | b |
| Treatment | c | d |

For such 2×2 data sets, in situations where the baseline varies (e.g., varying baseline regeneration across clutches), the commonly used measures of effect size are the RR,

$$RR = \frac{\left(\frac{\text{\# ephyrae that regenerate}}{\text{total \# ephyrae}}\right) \text{in treated group}}{\left(\frac{\text{\# ephyrae that regenerate}}{\text{total \# ephyrae}}\right) \text{in control group}} = \frac{\frac{c}{(c+d)}}{\frac{a}{(a+b)}}$$

and the OR,

$$OR = \frac{\left(\frac{\text{\# ephyrae that regenerate}}{\text{\# ephyrae that do not regenerate}}\right) \text{in treated group}}{\left(\frac{\text{\# ephyrae that regenerate}}{\text{\# ephyrae that do not regenerate}}\right) \text{in control group}} = \frac{\frac{c}{(c+d)}}{\frac{a}{(a+b)}}$$

RR compares the probability of an outcome in treated versus control group, whereas OR compares the odds of an outcome in treated versus control group.

For measurements of arm length and body size, the data sets are in the form of continuous variables. For such data, the commonly used effect size is the Response Ratio (R),

$$R = \frac{\text{mean arm length in treated group}}{\text{mean arm length in control group}}$$

R evaluates the proportionate change that results from a treatment, and is the meaningful effect size to use when the outcome of a treatment is measured on a physical scale, for example, length or area (as opposed to arbitrary scale, e.g., happiness level). Experiments where regeneration in one of the groups occurred in 0 ephyra were necessarily excluded.

Having computed the effect size (RR, OR, or R) within each experiment, meta-analysis of the effect size across experiments was performed. The metafor package (*Viechtbauer, 2010*) in R was used, with fixed-effect model (for nutrients and leucine) or random-effect restricted maximum likelihood model (for insulin and hypoxia, which had different control conditions across the experiments). Statistical coefficients were based on normal distribution.

## Phalloidin and tyrosinated tubulin staining

All steps were performed at room temperature, unless indicated otherwise. Ephyrae were first anesthetized in 400 µM menthol, which minimizes curling during fixing. Next, ephyrae were fixed in 3.7% (v/v) formaldehyde (in phosphate-buffered saline [PBS]) for 15 min, permeabilized in 0.5% Triton X-100 (in PBS) for 5 min, and blocked in 3% (w/v) BSA for 2 min. For neuron staining, ephyrae were incubated in 1:200 mouse anti-tyrosinated alpha tubulin antibody (Sigma-Aldrich MAB1864-I) overnight at 4°C, and then in 1:200 goat-anti-mouse Alexa Fluor 488 (Life Technologies A11029) overnight in the dark at 4°C. Primary or secondary antibodies were diluted in 3% BSA. For actin staining, ephyrae were incubated in 1:20 Alexa Fluor 555 Phalloidin (Life Technologies A12379) overnight or for 2 hr in the dark at 4°C. For nuclei staining, ephyrae were incubated in 1:10 Hoechst 33342 (Sigma-Aldrich B2261) for 30 min in the dark.

## Microscopy

Ephyrae were imaged anesthetized in menthol. Brightfield images, fluorescent images, and movies were taken with the Zeiss AxioZoom.V16 stereo zoom microscope and AxioCam HR 13-megapixel camera. Optical sectioning was performed with ApoTome.2.

## *Drosophila melanogaster*

CantonS wild-type strain was a gift from Peter Lee in Kai Zinn's lab at Caltech. OregonR was a gift from James McGehee in Angela Stathopolous' lab at Caltech. OregonR and CantonS flies were reared under standard conditions at 23°C, sometime supplemented with baker's yeast.

## Regeneration experiments

Amputation was performed on adult flies 2–7 days after eclosion. Flies were anesthetized with $CO_2$, placed under a dissection microscope, and tibia amputated using a spring scissors (Fine Science Tools, 91500-09) and superfine dissecting forceps (VWR, 82027-402). See *Figure 4* for detailed description of the amputation plane. Recovering *Drosophila* were allocated randomly to vials with standard lab fly food (control) or standard lab fly food mixed with 1.7 mM L-Leucine (Sigma-Aldrich L8000), 1.7

mM L-Glutamine (Sigma-Aldrich G3126), and 33 µg/ml insulin (human recombinant, MP Biomedicals 0219390080). To introduce the nutrient supplements, the fly food was microwaved in short pulses, such that the topmost layer of the food was liquified. The supplements in aqueous stocks were then pipetted into this liquified layer. Food was allowed to re-set at 4°C for at least 20 min. New food was prepared fresh every 2 days, and flies were moved into freshly prepared treated food every 2 days, throughout the course of the 2- to 3-week experiment.

## Single-fly tracking

Amputation and treatment were performed as in the bulk regeneration experiments described above, with the following modifications. Canton S flies were amputated in different limbs, and were housed in small groups such that in any given vial, each fly was uniquely identifiable by sex and amputated limb. Typically, 1–6 flies are housed in each vial. 3–5 days treatment produced similar response as sustained treatment; therefore, for simplicity, treatment was performed for 3–5 days. Treated flies, but not control flies, were sometime continued on yeast supplementation. Flies were imaged immediately after amputation and 1–3 additional times over the course of 2–4 weeks. As anesthetized flies jitter, images were taken in a video format, and single frames were then selected for analysis in which the leg stump was in focus. Tibia length was quantified in ImageJ as the diameter of the minimum enclosing circle of the leg to achieve rotation-invariant assessments. Identical imaging and analysis procedures were used for treated and control flies.

## Statistical analysis and blind

Blind measurements were performed on one pair of control and treated datasets. Statistical comparison of percent change in length in control and treated leg stumps was performed using the non-parametric Kruskal-Wallis test. Samples were assigned integer ranks from smallest to largest by quantity, and then differences between conditions were assessed with respect to the ranks. The p-value tests the null hypothesis that the data are drawn from the same distribution.

## DAPI staining

Fly tibias were dissected and washed in 70% ethanol (<1 min) to decrease the hydrophobicity of the cuticle and washed in PBS with 0.3% Triton-X for 10 min. The legs were fixed in 4% paraformaldehyde (in PBS) overnight at 4°C and washed five times for 20 min each in PBS with 0.3% Triton-X. The legs were equilibrated in Vectashield mounting medium with DAPI (Vector H-1200) overnight at 4°C, and imaged using Zeiss AxioZoom.V16 stereo zoom microscope with AxioCam HR 13-megapixel camera. Confocal imaging was performed using X-Light V2 spinning disk mounted on the Olympus IX81 inverted microscope.

## Live-fly imaging

Flies anesthetized on a $CO_2$ bed were imaged under a Zeiss SteREO Discovery.V8 stereomicroscope equipped with the Zeiss AxioCam 503 color camera.

## Electron microscopy

Environmental scanning electron microscopy (ESEM) was performed on a FEI Quanta 200F (FEI, Hillsboro, OR). Whole-live flies were mounted onto the SEM stub with copper tape. ESEM images were attained at a pressure of 0.1 mbar and 5 kV at a working distance of 9–12 mm, with water as the ionizing gas.

## *Mus musculus*

All studies comply with relevant ethical regulations for animal testing and research, and received ethical approval by the Institutional Animal Care and Use Committees at the California Institute of Technology.

## Strain

Adult female (3–6 months old) wild-type CD1 mice (Charles River Laboratories strain 022) were used for all regeneration studies.

## Regeneration experiments

Digit amputation was performed following the established protocol in the field (*Simkin et al., 2013*). Mice were anesthetized with 1–5% isoflurane (in oxygen) in an induction chamber, followed by maintenance on a nosecone. The mouse was positioned on its belly with its hind paws outstretched and the ventral side of the paw facing upwards. Sustained-Release Buprenorphine was administered (Buprenorphine SR LAB) at 0.5 mg/kg subcutaneously as an analgesic. Blood flow to the hindlimb was stemmed by tying a rubber band around the ankle and clamping it with a hemostat. All surgical procedures were carried out under a Zeiss Stemi 305 dissection microscope. An initial incision, parallel to the position of foot, was made through the ventral fat pad using Vannas spring scissors (World Precision Instruments, 14003). The length of this incision was determined by the amount of ventral skin needed to seal the digit amputation wound completely. The ventral skin freed in the initial incision was peeled back using surgical forceps, and a no. 10 scalpel (Sklar, 06-3110) was used to amputate and bisect the digit completely through the second or third phalange. Digits 2 and 4 on the right hind paw were operated on in this fashion, while digit three remained unamputated as an internal control. The amputation wound was immediately closed with the ventral skin flap and sealed with GLUture (Zoetis, Kalamazoo, MI). Amputated portions were immediately fixed as control for skeletal staining. Dissolved 1.5% L-leucine (USP grade, VWR E811), 1.5% L-glutamine (USP grade, Sigma-Aldrich G8540), and 4–10% sucrose (AR ACS grade, Avantor 8360) in drinking water was administered to mice in the experimental group ad libitum after amputation. Control mice were given untreated drinking water. Drinking water was refreshed weekly for both control and experimental groups, and treated water was made fresh on the day that drinking water was replaced. The amputated digit stumps were photographed weekly for 7–8 weeks, at which time the digits were dissected for skeletal staining.

## Statistical analysis

The sample size in the experiment balanced the aim of achieving >90% confidence level with ethical consideration of minimizing the number of animals used. Animals were randomly allocated to the control or treatment group. No restricted randomization was applied. For weight measurement, the unit of analysis is a single animal. For regeneration phenotype, the unit of analysis is a single digit. Student's t-test was used to evaluate the null hypothesis that there is no difference between the control and treated groups. 95% CIs were computed assuming normal distribution. All data were included in the analysis.

## Mouse digit dissection and skeletal staining

Mice were euthanized and digits 2, 3, and 4 were removed with a no. 10 scalpel (Sklar, 06-3110) through the first phalange. Excess skin and flesh were removed with spring scissors (Fine Science Tools, 91500-09) and fine dissecting forceps (Fine Science Tools, 11254-20). All digits analyzed by whole-mount skeletal stains were prepared with a standard alizarin red and alcian blue staining protocol (*McLeod, 1980*). Digits were dehydrated in 95% ethanol for 1 day, and incubated in staining solution (0.005% alizarin red (Beantown Chemical, BT144735), 0.015% alcian blue (Acros Organics, AC40046-0100), 5 % acetic acid, 60 % ethanol) for 1 day at 37 °C. Tissue was cleared in 2 % potassium hydroxide at room temperature for 1 day, 1% potassium hydroxide for 1 day, and then taken through an increasing glycerol series (25%, 50%, 75%, and 100%). The stained samples were imaged on Zeiss AxioZoom.V16 stereo zoom microscope with a Zeiss AxioCam 503 color camera or a Zeiss Stemi 305 dissection microscope with an iPhone six camera.

## Acknowledgements

The authors thank Kiersten Darrow and Michael Schaadt at the Cabrillo Marine Aquarium, Cabrillo, CA for the gifts of Aurelia polyps and help with jellyfish culture, Patrick Leahy at the Caltech's Kerckhoff Marine Laboratory, Corona Del Mar, CA for the help with setting up Aurelia experiments in natural habitat, Matthew Hunt from the Kavli Nanoscience Institute at Caltech for help with SEM imaging, James McGehee, Angela Stathopolous, Peter Lee, Kai Zinn for sharing *Drosophila* strains, Yujing Yang and Long Cai for help with imaging, and Carlos Lois for demonstration of digit amputation. The authors thank Gertrud Schupbach, Natalie Andrew, and Aki Ohdera for comments on the manuscript; and Katalina Fejes-Toth, Luis del Peso, Aditya Saxena, and Marta Truchado Garcia for

discussion. This work was supported by the National Science Foundation Graduate Research Fellowship Program (1144469; to MJA), the James E and Charlotte Fedde Cordes Postdoctoral Fellowship in Biology (to DAG); the James S McDonnell Foundation for Complex Systems Science (220020365; to LG), the Center for Environmental Microbial Interactions at Caltech (to ZC and LG), the Center of Evolutionary Sciences at Caltech (to YL and LG), the Summer Undergraduate Research Fellowships (to IL and LG), and Charles Trimble and Caltech's Biology and Biological Engineering Chair's Council Inducing Regeneration Fund (to LG).

## Additional information

### Competing interests

Michael J Abrams, Fayth Hui Tan, Ty Basinger, Martin L Heithe, Lea Goentoro: Inventor in patent rights application for findings in the manuscript. The other authors declare that no competing interests exist.

### Funding

| Funder | Grant reference number | Author |
|---|---|---|
| James S. McDonnell Foundation | Complex Systems Science | Lea Goentoro |
| National Science Foundation | Graduate Research Fellowship | Michael J Abrams |
| James E. and Charlotte Fedde Cordes | Postdoctoral Fellowship | David A Gold |
| Caltech Associates | Environmental Microbial Interactions | Zevin J Condiotte David A Gold Lea Goentoro |
| Charles Trimble and Caltech's Biology and Biological Chair's Council Inducing Regeneration Fund | | Lea Goentoro |
| Caltech Associates | Evolutionary Sciences | Yutian Li Lea Goentoro |
| National Institute of Standards and Technology | SURF | Iris T Lee Lea Goentoro |
| James S. McDonnell Foundation | 220020365 | Lea Goentoro |
| National Science Foundation | 1144469 | Michael J Abrams |

The funders had no role in study design, data collection and interpretation, or the decision to submit the work for publication.

### Author contributions

Michael J Abrams, Conceptualization, Data curation, Formal analysis, Investigation, Methodology, Software, Validation, Visualization, Writing – review and editing; Fayth Hui Tan, Conceptualization, Data curation, Formal analysis, Investigation, Methodology, Validation, Visualization, Writing – review and editing; Yutian Li, Conceptualization, Data curation, Investigation, Methodology, Validation, Visualization, Writing – review and editing; Ty Basinger, Iris T Lee, Zevin J Condiotte, Data curation, Investigation, Methodology, Validation, Writing – review and editing; Martin L Heithe, Data curation, Formal analysis, Investigation, Methodology, Validation, Writing – review and editing; Anish Sarma, Data curation, Formal analysis, Investigation, Methodology, Software, Validation, Writing – review and editing; Misha Raffiee, Investigation, Writing – review and editing; John O Dabiri, Supervision, Writing – review and editing; David A Gold, Investigation, Methodology, Writing – review and editing; Lea Goentoro, Conceptualization, Data curation, Formal analysis, Funding acquisition, Investigation, Methodology,

Project administration, Resources, Software, Supervision, Validation, Visualization, Writing – original draft

#### Author ORCIDs
Yutian Li http://orcid.org/0000-0001-8151-3518
Lea Goentoro http://orcid.org/0000-0002-3904-0195

#### Ethics
All studies comply with relevant ethical regulations for animal testing and research, and received ethical approval by the Institutional Animal Care and Use Committees (IACUC) at the California Institute of Technology. The protocol was approved by the IACUC at Caltech under the protocol number 1773-19 . All mouse surgery was performed under isoflurane anesthesia, and every effort was made to minimize suffering.

#### Decision letter and Author response
Decision letter https://doi.org/10.7554/eLife.65092.sa1
Author response https://doi.org/10.7554/eLife.65092.sa2

---

## Additional files

#### Supplementary files
• Transparent reporting form

#### Data availability
All data generated or analysed during this study are included in the manuscript and supporting files, as well as deposited to the open repository CaltechDATA.

The following datasets were generated:

| Author(s) | Year | Dataset title | Dataset URL | Database and Identifier |
|---|---|---|---|---|
| Abrams MJ, Basinger T, Goentoro L | 2021 | Regeneration data - Jellyfish | https://doi.org/10.22002/D1.2076 | CaltechDATA, 10.22002/D1.2076 |
| Li Y, Sarma A, Lee IT, Condiotte Z, Goentoro L | 2021 | Regeneration data - *Drosophila* | https://doi.org/10.22002/D1.2157 | CaltechDATA, 10.22002/D1.2157 |
| Tan FH, Heithe ML, Goentoro L | 2020 | Regeneration data - mouse | https://doi.org/10.22002/D1.1790 | CaltechDATA, 10.22002/D1.1790 |

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
