## [Editor Report]

This paper shows that simple nutritional interventions such as L-leucine, insulin or sucrose can trigger appendage regeneration in three species that do not regenerate appendage in normal conditions, the Aurelia jellyfish, *Drosophila* flies and mice. The results are stunning and provide novel model systems to induce appendage regeneration in animals and to study the mechanisms underlying regeneration.

---

## [Decision Letter]

**Decision letter after peer review:**

Thank you for submitting your article "A conserved strategy for inducing appendage regeneration" for consideration by *eLife*. Your article has been reviewed by 3 peer reviewers, including Virginie Courtier-Orgogozo as the Reviewing Editor and Reviewer #1, and the evaluation has been overseen by Patricia Wittkopp as the Senior Editor. The following individual involved in review of your submission has agreed to reveal their identity: Eve Gazave (Reviewer #3).

Essential revisions:

(1) Prior to this study, *Drosophila* adult legs were not thought to be capable of any regeneration. Given the low number of legs on which observations were made (apparently only 7 legs) and the authors' mention of removing contaminating flies with 'uncut tibias or wrong cuts', better documentation is needed to be fully convinced that these few cases represent genuine instances of regeneration. See detailed comment of Reviewer #2 below regarding this important point.

If such data are impossible for you to obtain at this point, then we would suggest to remove the part on *Drosophila* from your paper and focus your manuscript on Aurelia and mice only. Removing the *Drosophila* analysis would significantly reduce the broadness of your study, but it would still be considered for publication in *eLife*. We would much prefer that you include additional data and confirm your *Drosophila* results, as this would have much more impact in the field. Furthermore, we greatly acknowledge the original approach of the authors to use three different models to deal with a biological question in a comparative manner. Removing one of the three models would greatly diminish this aspect.

(2) Currently figure 6 presents a few selected examples which are not easy to compare between treated mice and control groups because they have variable or unspecified sites of amputation (except panels 6i and k, which are comparable). It would be good to compare more carefully the treated animals and controls with respect to the precise position of the amputation site.

(3) The claim that treatments 'induce' regeneration seems exaggerated, given that appendage regeneration in Aurelia and in mouse digits can occur to a variable extent in untreated animals. These treatments appear to increase the frequency and extent of regeneration in the context of a limited and variable regenerative response. This should be rephrased in the text.

(4) Please justify why in Aurelia you performed an amputation that remove almost the half body of the ephyra and not just one arm.

(5) Can you please also comment on the high variability observed in Aurelia, including the regenerative response in the controls.

(6) Please take into account the various comments below regarding the description of the methods and the presentation of the results.

(7) An interesting paper (Williams et al. 2021 Dev biol, https://www.sciencedirect.com/science/article/pii/S0012160621000166) was published recently about nutrient availability and regeneration ability in *Xenopus*. *Xenopus* usually lose regenerative competence at a certain stage and the authors were able to restore this regenerative competence by abundant feeding. This study reinforces the idea of the present paper, so maybe it could be mentioned.

*Reviewer #1 (Recommendations for the authors):*

Figure 3c-f: explain in the legend what "Exp", "N" and the various numbers mean.

Line 187: "bovine serum albumin at the same molarity showed no effect". Where is this data shown? How many individuals were tested? The Material and Methods mentions that BSA was used at 500nM and 3nM.

Line 193: "reducing oxygen using argon flow similarly increased regeneration frequency". Where is this data shown?

Figure 3-Figure supplement 1. "Various physical environments for the ephyrae recovering from injury were tested, e.g., shallow vs deep water, seawater with varying salinity, cold vs warm temperature, light versus dark, stagnant water vs current, generating water current through various means, including shaking or rotating to generate turbulent mixing and as shown here air bubbling a conical tube to generate vertical current". Where is this data shown?

Figure 3-Figure supplement 2. The colors black/red of the legend box should be inverted.

Figure 3-Figure supplement 2 legend: typo in "These trees verify the he simple trees are".

Figure 4C. What is the black line on the femur after amputation? Is it an artefact/dust? If so, it would be better to show another leg.

Line 302: among the 925 control flies raised in normal food what was the percentage of white-tip tibia? This is not clear from the text. Also, it would be good to mention which strain(s) was used for normal food conditions.

In Aurelia and mice several organs in the same animal were amputated and examined a few days later. Is there a correlation within a given animal, i.e. is it more likely to find a regenerated organ when the next one is regenerated in the high-leucine-high-sugar/insulin condition?

Was leg amputation in *Drosophila* performed on males or females or on a mix of both sexes?

It is very nice that raw image data of ephyrae and mouse digits were deposited in the open repository CaltechDATA. Could the authors also deposit their raw images of *Drosophila* legs?

*Reviewer #2 (Recommendations for the authors):*

1) Under specific conditions the frequency and extent of arm regeneration in Aurelia increases compared to control conditions. The data on Aurelia are convincing but we would urge the authors to be more explicit about arm regeneration in control conditions. The data presented in figure 3c-f and in the supplementary materials suggest that the frequency of arm regeneration is highly variable in control conditions (ranging from 0 to 85%). While the conclusion that high-nutrient, insulin, hypoxia and leucine treatments increase the frequency or extent of regeneration seems justified, the suggestion that these treatments 'induce' regeneration is somewhat misleading. The data suggest that these treatments increase the frequency and extent of regeneration in the context of a limited and variable regenerative response.

As a side point, it would be interesting to hear if the authors have any ideas on the source of this variability. Could it be explained by stochastic differences in nutrition, under the nutrient-limiting conditions in the control treatment?

2) The observation of leg regeneration in adult *Drosophila* is, to our knowledge, unprecedented and very intriguing. Given the low number of legs on which these observations were made (apparently only 7 legs), we would need better documentation to be fully convinced that these represent genuine instances of regeneration. In particular:

a) The amputation site on the tibia is indicated in figure 4c, but it is unclear how rigorously the cuts were made and tracked during these experiments (the authors mention removing contaminating flies with 'uncut tibias or wrong cuts', suggesting that amputation errors and contamination did occur). We would recommend marking the amputation site (e.g. using a permanent dye) to make sure that the extent of new growth from the cut site can be properly assessed in each individual. In the current data it is not possible to verify which was the site of amputation in each of the regenerated limbs.

b) In order to understand how a patterned leg can re-grow and emerge with a normal cuticle in the absence of molting, it will be necessary to document the process of growth, rather than just the final outcome. For example, it is currently not clear to what extent the images shown in figure 5e might represent different stages of regeneration, rather than variable endpoints. Documenting the timecourse of regeneration on individual limbs would confirm regrowth from the amputated limb stump and show how patterned leg tissue can emerge in the absence of molting. This should be straightforward to do by anaesthetising the flies and imaging the legs at regular intervals under a stereoscope.

c) A variety of genetically-encoded fluorescent markers widely available in *Drosophila* (e.g. markers for nuclei, epithelial cell outlines, cell proliferation) could be used to describe further this process of growth and regeneration.

3) The experiments on mouse digits suggest that combined supplementation of leucine, glutamine and sucrose influence the likelihood of digit regeneration, when amputations are performed near a critical boundary at the base of the nail. As shown in figure 6d, the sites of amputation varied, within a region that is known to contain a critical boundary for the ability to regenerate digits (distal cuts regenerate, proximal cuts do not). To be confident of this effect, we feel that the treated animals and controls should be compared more carefully with respect to the precise position of the amputation site. Currently figure 6 presents a few selected examples which are not easy to compare between treated and control groups because they have variable or unspecified sites of amputation (except panels 6i and k, which are comparable). The tables presented in supplementary figure 6 make a useful distinction between amputations in P2 or P3, but the images are not easy to relate to the data presented on that table. The authors also make a useful distinction between patterned and unpatterned responses, but again, it is unclear how these responses relate to the site of amputation and the data presented on the table (it seems there were only 2 cases of patterned regeneration).

Given that digit regeneration is possible when digits are amputated in a distal location, we think it is somewhat misleading to regard the nutritional treatment as an 'inducer' in a system that is normally incapable of regeneration. Rather, we suggest regarding the treatment as a condition that modulates the frequency and extent of regeneration in a system that has a limited capacity to regenerate. In this respect, the nutritional effects uncovered by this study are interesting variables to consider along with other factors, such as amputation site, age and mouse strain.

4) We would urge the authors to be more cautious and nuanced on some of the conclusions drawn from these results. Specifically:

The authors suggest that the effects of these nutritional interventions 'blur the boundary between regenerating versus non-regenerating animals' and suggest that evolutionarily dormant modules for regeneration could be activated in non-regenerating animals (e.g. lines 457-458 and 469). As discussed earlier, it seems that the arm regeneration in Aurelia and mice extends an existing, limited regenerative capacity; no 'evolutionary dormant' modules need to be invoked, as they are already (conditionally) active in these species. The case of *Drosophila* is more relevant for this argument, but should be documented more firmly before such conclusions can be drawn.

The authors suggest that the same factors promote regeneration in diverse species, but the actual treatments applied in each species differ: in Aurelia high-nutrient, insulin and leucine treatments are tested separately, in *Drosophila* leucine and insulin are combined (with the addition of glutamine), in the mouse leucine and sucrose were combined (with glutamine). The authors explain some of these differences in experimental design, but these explanations are not all entirely convincing (e.g. insulin does not need to be administered orally to the mice). More importantly, the effects of these treatments are not identical as the authors suggest. Sucrose metabolism and insulin are linked, but one provides a caloric input while the other elicits a hormonal response. Human insulin fed to *Drosophila* will not have identical effects to the addition of sugar (insulin-like peptides are known to mediate feeding preferences and growth control in *Drosophila*). We accept the suggestion that these interventions could have similar effects on metabolism, but we would urge the authors to be more cautious in their interpretation and more explicit on the potential differences of the treatments they applied in different species.

*Reviewer #3 (Recommendations for the authors):*

Introduction:

Please provide references, related to the old studies you mentioned (lines 32 and so on).

In addition to the conserved regeneration-responsive enhancers identified recently, other studies support the idea of some molecular conservation in regeneration (ex Cary et al. 2019). Please add this point and associated references.

L48 to 60. Frog and mice harbor some relative regenerative capabilities depending on age. Please precise at which stage/ age the experiments were done (ie for the mouse digit example, is it on neonatal mice?)

L63: "Regeneration tends to decrease with age" this is essentially true for vertebrates, many "invertebrates" can regenerate all lifelong.

L65-68: Please add relevant references, associated to continuous growth in annelids.

Results:

Concerning Aurelia, the data shown are highly variable from one experiment to another. Concerning the *Drosophila* experiments, around 1% of *Drosophila* regrow a tibia. To support these results and to understand such variability, it would be great to have RNA seq data on controls versus treated individuals able to regenerate versus treated individuals that are not able to regenerate to approach the mechanistic underlying successfully induced regeneration.

L86: "it would more likely be intact" what do you mean here?

L89: Some medusa regenerate very well complex structures and organs, see old literature and recent paper Sinigaglia et al. 2020 e*Life*.

L113 and following: It is not clear for me the kind of amputation you did. Did you remove a single arm or more? Did you perform always the same king of amputation? If you amputate one or more arms is it changing the percentage of individuals able to regenerate?

L114: Why did you do your experiments in the original habitat?

L116: There is not figure 1e in the figure 1.

L121 and so on: How did you chose your "various molecular and physical factors"?

L125: Why did you amputate half of the ephyrae and not a single arm. Are you still in the context of appendage regeneration when removing half of the body?

L163: no effects observed by using modulators of developmental pathways. How was it done precisely?

L167: Is it unexpected to have water current necessity for an ephyra?

L183: Any dose effect for addition of insulin?

Figure 3: legends are missing to help the reader to better understand the figure.

L252: what about adding leucine to HF?

L279: why not testing leucin and insulin alone before adding both in the *Drosophila* food?

Discussion

L451: I am not sure to understand how the fact that regenerations at the metazoan scale rely on either stem cells or differentiated cells is supporting the idea of commonality and ancestry and upstream shared regulation.

L456: Is there any previous study showing the importance of nutrient for regeneration?

L457: Do we have insight about the importance of environment for regeneration in some model species?

L465 to 470: I am not sure to follow this part of the discussion.

L472: Importance of food availability and so nutrient dependence have been described for ctenophore a highly regenerating model. see Bading et al. 2017. I am not

Is there any previous data, such as RNA-seq data of regeneration that may point out on a role of energy/nutrient etc on regeneration?

[Editors' note: further revisions were suggested prior to acceptance, as described below.]

Thank you for submitting your article "A conserved strategy for inducing appendage regeneration" for consideration by *eLife*. Your revised article has been reviewed by a Reviewing Editor and Patricia Wittkopp as the Senior Editor.

We appreciate that you increased the sample size for experiments involving *Drosophila*. This has greatly strengthened your study. The revisions below would help to further increase the clarity of the manuscript.

Essential revisions:

1) Line 47: "Although the notion of convergences still cannot be fully excluded (e.g., Lai and Aboobaker, 2018), these findings begin to build the case that, rather than many instances of convergence, the ability to regenerate is ancestral (Sanchez-Alvarado, 2000; Bely and Nyberg, 2010)".

We would suggest to replace it with "Although the hypothesis of convergent evolution cannot be fully excluded (e.g., Lai and Aboobaker, 2018), these findings begin to build the case that the ability to regenerate may be ancestral (Sanchez-Alvarado, 2000; Bely and Nyberg, 2010).

2) Line 89-91: The following sentence is difficult to understand: "There is no guarantee that regeneration can be induced, e.g., because the organ never evolved the ability to regenerate or because regeneration that has been inactivated could be rendered vestigial over time." We would suggest to remove it and change in the next sentence: "it would more likely be intact" by "it would likely remain intact".

3) Line 122: replace "a small bud would appear at the amputation site" by "a small bud at the amputation site".

4) Line 324-325: "the tips stained positively with DAPI instead": what is the proportion of tips that stained positively? How many tips were examined in total?

5) Line 359-360: "5 flies showed reformed tibia segments". Please indicate again the total number of flies that were examined.

6) Line 706: "Treated flies were supplemented with yeast." And what about the control flies?

7) Line 991: " Experiments were performed at 68°F". Please use °C.

---

## [Author Response]

Essential revisions:(1) Prior to this study, *Drosophila* adult legs were not thought to be capable of any regeneration. Given the low number of legs on which observations were made (apparently only 7 legs) and the authors' mention of removing contaminating flies with 'uncut tibias or wrong cuts', better documentation is needed to be fully convinced that these few cases represent genuine instances of regeneration. See detailed comment of Reviewer #2 below regarding this important point.If such data are impossible for you to obtain at this point, then we would suggest to remove the part on Drosophila from your paper and focus your manuscript on Aurelia and mice only. Removing the *Drosophila* analysis would significantly reduce the broadness of your study, but it would still be considered for publication in eLife. We would much prefer that you include additional data and confirm your Drosophila results, as this would have much more impact in the field. Furthermore, we greatly acknowledge the original approach of the authors to use three different models to deal with a biological question in a comparative manner. Removing one of the three models would greatly diminish this aspect.

We sincerely thank the Editor and Reviewers for their suggestions, which we believe have strengthened the manuscript. The revised manuscript now includes quantitative single-fly tracking experiments that rigorously document the regenerative response in individual flies. The new results are included in the main text (lines 333-361 and revised Figure 5). The original fly section (lines 287-332 and Figure 4) has been made more concise to accommodate the new results.

To track regenerative response in individual flies, we housed in each vial a small number of flies that were amputated in different limbs (Figure 5a). The amputated limb position, combined with sex, enabled unique identification of each fly within a vial. Each fly was imaged immediately after amputation and 1-3 additional times over the course of 2-4 weeks. The imaging frequency balanced obtaining time-lapse information with minimizing stress from repeated anesthesia.

While the previous bulk experiments captured the dramatic phenotype (i.e., a completed leg segment), the quantitative single-fly tracking enabled capturing various extents of leg growth. Time-lapse pictures of control and treated flies are shown in Figure 5b-d. Control tibia stumps showed near-zero percent change in length (Figure 5d; mean -0.3%, 95% CI [-3.8, 3.2%], N=116). By contrast, 49% of treated flies showed growth beyond the 95% confidence intervals of the control distribution (Figure 5f; N=150). The single-fly tracking also enabled capturing intermediate morphologies. For instance, some new growths are pigmented differently and have no sensory bristles (Figure 5g); showed white tissues protruding from the end (Figure 5—figure supplement 1c), which in some stumps is remodeled over time (Figure 5—figure supplement 1d). We believe that results from the *Drosophila* experiments now demonstrate that patterned regrowth can be induced from adult fly limb. The source file “*Drosophila* data.xls” contains the raw data for generating Figure 5e-f. Raw image files are being deposited in the open repository CaltechDATA, doi.org/10.22002/D1.2157.

(2) Currently figure 6 presents a few selected examples which are not easy to compare between treated mice and control groups because they have variable or unspecified sites of amputation (except panels 6i and k, which are comparable). It would be good to compare more carefully the treated animals and controls with respect to the precise position of the amputation site.

We presented the direct comparison of amputation sites in the tables in Figure 6—figure supplement 1. To follow the reviewers’ suggestion and make the information more visual, we revised Figure 6 to include P3 amputation control. The different lighting was due to the different microscopes accessible during the pandemic, but the morphologies are clearly comparable across all images. Finally, we added for the revision a source file “Mouse data.xls” spreadsheet which lists for each digit analyzed the amputation type, the phenotype score, and the corresponding raw image filename of the bone staining. The raw image files have been deposited to CaltechDATA, doi.org/10.22002/D1.1790.

(3) The claim that treatments 'induce' regeneration seems exaggerated, given that appendage regeneration in Aurelia and in mouse digits can occur to a variable extent in untreated animals. These treatments appear to increase the frequency and extent of regeneration in the context of a limited and variable regenerative response. This should be rephrased in the text.

In *Aurelia* where partial regeneration variably occurs in the natural habitat, we agree with the Reviewer and have systematically gone through the *Aurelia* section to ensure the findings are described as “increasing the frequency of regeneration” or “promote regeneration”.

In mice, however, proximal mouse digits do not regenerate—not at any age and not in any wild strains. Our wording is in accordance to the literature in the field (e.g., PMID 23760480, PMID 20110320, PMID 30723209, PMID 33937937, PMID 29291973). We believe therefore in this case, our choice of wording is validated by the scientific context and precedents in the mouse digit field.

(4) Please justify why in Aurelia you performed an amputation that remove almost the half body of the ephyra and not just one arm.

We have explored various amputations schemes, e.g., removing 1, 2, 3, and 4 arms (PMID: 26080418). The amputation scheme used was chosen because it could be performed with the most ease, hence facilitating screening thousands of ephyrae. We added this rationale in Methods (lines 614-620). With the amputation scheme chosen, Reviewer 3 is correct that although arm regeneration is the most obvious outcome, body parts are also regenerated**—**thus broadening the possible implications of the study. We added this point in Discussion (lines 502-505):

“Notably in *Aurelia*, the amputation bisected through the body, and more than appendage was in fact regenerated, e.g., circular muscle in the body is regrown (Figure 2e). Thus, nutrient supplementation may have regenerative effect in body parts beyond appendage.”

(5) Can you please also comment on the high variability observed in Aurelia, including the regenerative response in the controls.

The variable regeneration is intriguing to us too, and we are actively probing for individual differences that correlate with regenerative response. We think the variability is a feature of triggering a latent process, which because it has been latent might have lost some robust connectivities and therefore surfaces in a way that is sensitive to environmental and physiological parameters (Discussion, lines 527-536, included below). It is interesting to us that *Aurelia* spontaneously show variable and incomplete regeneration in the natural habitat, suggesting to us leakiness in the linkage.

“… the regenerative response was unusually variable. The variability stands in stark contrast to the robust regeneration in e.g., axolotl, planaria, or hydra. […] It would be interesting next to identify individual differences that contribute to varying propensities to mount regenerative response.”

(6) Please take into account the various comments below regarding the description of the methods and the presentation of the results.

We have revised the manuscript accordingly. Thank you for all the suggestions.

(7) An interesting paper (Williams et al. 2021 Dev biol, https://www.sciencedirect.com/science/article/pii/S0012160621000166) was published recently about nutrient availability and regeneration ability in *Xenopus. Xenopus* usually lose regenerative competence at a certain stage and the authors were able to restore this regenerative competence by abundant feeding. This study reinforces the idea of the present paper, so maybe it could be mentioned.

We learnt about the paper after the submission and were excited by the finding. Reference to the study is now included in the last paragraph of the Discussion section (lines 554-555):

“In conclusion, this study suggests that an inherent ability for appendage regeneration is retained in non-regenerating animals and can be unlocked with a conserved strategy. In line with our findings, it was recently reported that feeding restores tail regeneration during the refractory period in the *Xenopus tropicalis* tadpoles (Williams et al., 2021). …”

Reviewer #1 (Recommendations for the authors):Figure 3c-f: explain in the legend what "Exp", "N" and the various numbers mean.

We thank Reviewer 1 again for all the suggestions. A legend has been added to Figure 3.

Line 187: "bovine serum albumin at the same molarity showed no effect". Where is this data shown? How many individuals were tested? The Material and Methods mentions that BSA was used at 500nM and 3nM.Line 193: "reducing oxygen using argon flow similarly increased regeneration frequency". Where is this data shown?

The BSA and argon data are now included in Figure 3—figure supplement 4. For BSA control, the relevant control is the 500 nM, we have corrected the Methods.

Figure 3-Figure supplement 1. "Various physical environments for the ephyrae recovering from injury were tested, e.g., shallow vs deep water, seawater with varying salinity, cold vs warm temperature, light versus dark, stagnant water vs current, generating water current through various means, including shaking or rotating to generate turbulent mixing and as shown here air bubbling a conical tube to generate vertical current"Where is this data shown?

These factors were tested as part of the screen. They have now been added to the source file ‘Aurelia.xls’ (Figure 2—source data 1).

Figure 3-Figure supplement 2. The colors black/red of the legend box should be inverted.

This has been fixed.

Figure 3-Figure supplement 2 legend: typo in "These trees verify the he simple trees are"

The typo has been fixed, thank you.

Figure 4C. What is the black line on the femur after amputation? Is it an artefact/dust? If so, it would be better to show another leg.

The black line was an air bubble. These images now got edited out to simplify Figure 4 and to incorporate new data on the single-fly tracking in Figure 5.

Line 302: among the 925 control flies raised in normal food what was the percentage of white-tip tibia? This is not clear from the text. Also, it would be good to mention which strain(s) was used for normal food conditions.

We agree it got confusing to look for this information just from the text. We now include a table to summarize the bulk data in Figure 4—figure supplement 1.

In Aurelia and mice several organs in the same animal were amputated and examined a few days later. Is there a correlation within a given animal, i.e. is it more likely to find a regenerated organ when the next one is regenerated in the high-leucine-high-sugar/insulin condition?

We are also interested in this question, and testing it by amputating different organs in the same animal.

Was leg amputation in *Drosophila* performed on males or females or on a mix of both sexes?

We now have data on this from the single-fly tracking experiment. No sex-based differences were apparent in the frequency of regenerative response (Figure 5—figure supplement 1).

It is nice that raw image data of ephyrae and mouse digits were deposited in the open repository CaltechDATA. Could the authors also deposit their raw images of *Drosophila* legs?

Previously with the bulk experiment, imaging was not performed for every fly. For the new single-fly tracking data, all videos analyzed are being uploaded in the open repository CaltechDATA, https://doi.org/10.22002/D1.2157. Because of the large size of the dataset—over 1000 files, each between 250 MB to 1 GB—the uploading was still ongoing at the time this revision was submitted; we will update the editorial office as soon as it is completed.

Reviewer #2 (Recommendations for the authors):1) Under specific conditions the frequency and extent of arm regeneration in Aurelia increases compared to control conditions. The data on Aurelia are convincing but we would urge the authors to be more explicit about arm regeneration in control conditions. The data presented in figure 3c-f and in the supplementary materials suggest that the frequency of arm regeneration is highly variable in control conditions (ranging from 0 to 85%). While the conclusion that high-nutrient, insulin, hypoxia and leucine treatments increase the frequency or extent of regeneration seems justified, the suggestion that these treatments 'induce' regeneration is somewhat misleading. The data suggest that these treatments increase the frequency and extent of regeneration in the context of a limited and variable regenerative response.

Previously with the bulk experiment, imaging was not performed for every fly. For the new single-fly tracking data, all videos analyzed are being uploaded in the open repository CaltechDATA, https://doi.org/10.22002/D1.2157. Because of the large size of the dataset—over 1000 files, each between 250 MB to 1 GB—the uploading was still ongoing at the time this revision was submitted; we will update the editorial office as soon as it is completed.

As a side point, it would be interesting to hear if the authors have any ideas on the source of this variability. Could it be explained by stochastic differences in nutrition, under the nutrient-limiting conditions in the control treatment?

Nutrient availability is not likely to be a major source of the variability as the low food condition is not close to a stochastic regime, at 100-200 rotifers/ephyra. Moreover, the variability was observed in both low and high food conditions. We find the variability intriguing too, and are actively probing for individual differences that correlate with regenerative response.

2) The observation of leg regeneration in adult *Drosophila* is, to our knowledge, unprecedented and very intriguing. Given the low number of legs on which these observations were made (apparently only 7 legs), we would need better documentation to be fully convinced that these represent genuine instances of regeneration. In particular:a) The amputation site on the tibia is indicated in figure 4c, but it is unclear how rigorously the cuts were made and tracked during these experiments (the authors mention removing contaminating flies with 'uncut tibias or wrong cuts', suggesting that amputation errors and contamination did occur). We would recommend marking the amputation site (e.g. using a permanent dye) to make sure that the extent of new growth from the cut site can be properly assessed in each individual. In the current data it is not possible to verify which was the site of amputation in each of the regenerated limbs.b) In order to understand how a patterned leg can re-grow and emerge with a normal cuticle in the absence of molting, it will be necessary to document the process of growth, rather than just the final outcome. For example, it is currently not clear to what extent the images shown in figure 5e might represent different stages of regeneration, rather than variable endpoints. Documenting the timecourse of regeneration on individual limbs would confirm regrowth from the amputated limb stump and show how patterned leg tissue can emerge in the absence of molting. This should be straightforward to do by anaesthetising the flies and imaging the legs at regular intervals under a stereoscope.c) A variety of genetically-encoded fluorescent markers widely available in *Drosophila* (e.g. markers for nuclei, epithelial cell outlines, cell proliferation) could be used to describe further this process of growth and regeneration.

We thank the Reviewer for the suggestions. We tested them, and marking the leg has not worked because the flies groom away the marking, while staining the fly legs is currently hampered by the high autofluorescence and low penetration of the cuticle. However, the single-fly tracking worked better than expected. The challenges to doing single-fly tracking were two-fold: (i) imaging the sub-millimeter-scale tibias in live flies is not straightforward as anesthetized flies jitter; (ii) stress due to repeated handling and anesthesia appears to reduce regenerative response. We systematically worked out these technical problems over these past months.

To track single flies, we housed in each vial a small number of flies that were amputated in different limbs (Figure 5a). The limb position combined with sex enabled unique identification of each fly in a vial. Each fly was videotaped immediately upon amputation, and subsequently 1-3 additional times over 2-4 weeks of the experiments. The imaging frequency balanced obtaining time-lapse information and minimizing stress from repeated anesthesia.

While the previous bulk experiments captured the dramatic phenotype with a completed leg segment, the single-fly tracking enabled capturing flies showing various extents of leg growth. Time-lapse pictures of control and treated flies are shown in Figure 5b-d. Control tibia stumps showed near-zero percent change in length (Figure 5d; mean -0.3%, 95% CI [-3.8, 3.2%], N=116). By contrast, 49% of treated flies showed growth beyond the 95% confidence intervals of the control distribution (Figure 5f; N=150). Moreover, as the Reviewer suggested, the single-fly tracking enabled capturing intermediate morphologies. For instance, some new growths are pigmented differently and have no sensory bristles (Figure 5g); showed white tissues protruding from the end (e.g., Figure 5—figure supplement 1c), which in some stumps is remodeled over time (e.g., Figure 5—figure supplement 1d).

The new results are included in the main text (lines 333-361 and revised Figure 5). The original fly section (lines 287-332 and Figure 4) has been made more concise to accommodate the new results. We believe that altogether, results from the *Drosophila* experiments now rigorously demonstrate that patterned regrowth can be induced from adult fly limb.

3) The experiments on mouse digits suggest that combined supplementation of leucine, glutamine and sucrose influence the likelihood of digit regeneration, when amputations are performed near a critical boundary at the base of the nail. As shown in figure 6d, the sites of amputation varied, within a region that is known to contain a critical boundary for the ability to regenerate digits (distal cuts regenerate, proximal cuts do not).

The boundary of the regenerating vs non-regenerating regions is well established (e.g., PMID: 28493324, PMID: 7100922, PMID: 18234177, PMID: 17147657). The regenerating tip is the third distal part *within* the nail organ. In our experiments, we amputated far from the regenerating region, *below* the base of the nail. Further, for each digit stump, the amputation plane was precisely determined by histological analysis of the digit part removed.

To be confident of this effect, we feel that the treated animals and controls should be compared more carefully with respect to the precise position of the amputation site. Currently figure 6 presents a few selected examples which are not easy to compare between treated and control groups because they have variable or unspecified sites of amputation (except panels 6i and k, which are comparable). The tables presented in supplementary figure 6 make a useful distinction between amputations in P2 or P3, but the images are not easy to relate to the data presented on that table. The authors also make a useful distinction between patterned and unpatterned responses, but again, it is unclear how these responses relate to the site of amputation and the data presented on the table (it seems there were only 2 cases of patterned regeneration).

The precise comparison was presented in Table in the supplementary figure 6. Six digits (12.5%) showed patterned regenerative response. To clarify the definition of amputation, revised Figure 6 now includes additional control images to clearly define P3 versus P2 amputations. The different lighting was due to the different microscopes we could access during the pandemic, but the morphologies can be clearly compared. Finally, we added for the revision a source spreadsheet file “Mouse data.xls” which lists for each digit analyzed the amputation type, the phenotype scoring, and the corresponding raw image file name. The raw image data have been deposited to CaltechDATA, doi.org/10.22002/D1.1790.

Given that digit regeneration is possible when digits are amputated in a distal location, we think it is somewhat misleading to regard the nutritional treatment as an 'inducer' in a system that is normally incapable of regeneration. Rather, we suggest regarding the treatment as a condition that modulates the frequency and extent of regeneration in a system that has a limited capacity to regenerate. In this respect, the nutritional effects uncovered by this study are interesting variables to consider along with other factors, such as amputation site, age and mouse strain.

Our choice of wording is in accordance to the literature in the mouse digit field (e.g., PMID: 23760480, PMID 20110320, PMID 30723209, PMID 33937937, PMID 29291973). To our knowledge, proximal mouse digits do not regenerate—not at any age and not in any wild strains; therefore there is no context here to say “modulating frequency”. It is also not necessarily correct or the convention to say that observing proximal response “extends” tip regeneration, because it is unclear the extent to which tip regeneration, which requires signals from the nail epithelium, involves different mechanisms (e.g., PMID: 23760480, PMID: 20110320). Thus, we believe therefore in this case, our choice of wording is validated by the scientific context and precedents in the mouse digit field.

4) We would urge the authors to be more cautious and nuanced on some of the conclusions drawn from these results. Specifically:The authors suggest that the effects of these nutritional interventions 'blur the boundary between regenerating versus non-regenerating animals' and suggest that evolutionarily dormant modules for regeneration could be activated in non-regenerating animals (e.g. lines 457-458 and 469). As discussed earlier, it seems that the arm regeneration in Aurelia and mice extends an existing, limited regenerative capacity; no 'evolutionary dormant' modules need to be invoked, as they are already (conditionally) active in these species. The case of *Drosophila* is more relevant for this argument, but should be documented more firmly before such conclusions can be drawn.

We agree with the Reviewer that it’s important to be cautious. The full version of the paragraph the Reviewer is referring to (now lines 518-536) explicitly lays out possible different interpretations of the data. Not all readers have to agree with the interpretation we favor, and we word our conclusion to provide space for differing ideas. We revise the paragraph to further clarify that we are describing interpretations:

“That regenerative response can be induced seemingly blurs the boundary between regenerating versus non-regenerating animals, because the factors identified in the study are not exotic: variations in amino acids, carbohydrates, and oxygen levels are conditions that the animals can plausibly encounter in nature. […] Alternatively, the interpretation we favor, what we observed is latent regeneration, which can be activated with broad environmental factors. … ”

The authors suggest that the same factors promote regeneration in diverse species, but the actual treatments applied in each species differ: in Aurelia high-nutrient, insulin and leucine treatments are tested separately, in *Drosophila* leucine and insulin are combined (with the addition of glutamine), in the mouse leucine and sucrose were combined (with glutamine). The authors explain some of these differences in experimental design, but these explanations are not all entirely convincing (e.g. insulin does not need to be administered orally to the mice). More importantly, the effects of these treatments are not identical as the authors suggest. Sucrose metabolism and insulin are linked, but one provides a caloric input while the other elicits a hormonal response. Human insulin fed to Drosophila will not have identical effects to the addition of sugar (insulin-like peptides are known to mediate feeding preferences and growth control in Drosophila). We accept the suggestion that these interventions could have similar effects on metabolism, but we would urge the authors to be more cautious in their interpretation and more explicit on the potential differences of the treatments they applied in different species.

We agree, and have included explicit discussion on the differences in the Discussion (lines 551-554):

“In conclusion, this study suggests that an inherent ability for appendage regeneration is retained in non-regenerating animals and can be unlocked with a conserved strategy. The treatments across species were not exactly identical, and correspondingly there might be differences in the precise molecular mechanisms – in spite of which the treatment worked across species in a predictive manner.”

Reviewer #3 (Recommendations for the authors):Introduction:Please provide references, related to the old studies you mentioned (lines 32 and so on).

The views of regeneration described are discussed in the three references that were cited on line 33 (the 1992 review by Richard Goss, the 1972 monograph by L.V. Polezhaev, and the 1901 book by Thomas Morgan). We repositioned these citations directly after the relevant sentences to make the source references clearer.

In addition to the conserved regeneration-responsive enhancers identified recently, other studies support the idea of some molecular conservation in regeneration (ex Cary et al. 2019). Please add this point and associated references.

This point and reference has been added (line 45):

“… even in poorly regenerative lineages, many embryonic and larval stages can regenerate. In regenerating animals, conserved molecular events (e.g., Cary et al., 2019, Kawakami et al., 2006) and regeneration-responsive enhancers (Wang et al., 2020) were identified. …”

L48 to 60. Frog and mice harbor some relative regenerative capabilities depending on age. Please precise at which stage/ age the experiments were done (ie for the mouse digit example, is it on neonatal mice?)

The specific study cited used neonatal and adult mice. We added this information to the sentence (L61). Proximal mouse digits do not regenerate in neonatal or adult mice.

“In neonatal and adult mouse digits, a model for exploring limb regeneration in mammals, bone outgrowth or joint-like structure can be induced via local implantation of Bmp2 (bone) or Bmp9 (joint; Yu et al., 2019).”

L63: "Regeneration tends to decrease with age" this is essentially true for vertebrates, many "invertebrates" can regenerate all lifelong.

Yes, thank you, this qualification has been added (L67):

“… regeneration especially in vertebrates tends to decrease with age, with juveniles and larvae more likely to regenerate than adults.”

L65-68: Please add relevant references, associated to continuous growth in annelids.

The reference has been added:

Rouse, G. (1998). "The Annelida and their close relatives". In Anderson, D. T. (ed.). Invertebrate Zoology. Oxford University Press.

Results:Concerning Aurelia, the data shown are highly variable from one experiment to another. Concerning the Drosophila experiments, around 1% of *Drosophila* regrow a tibia. To support these results and to understand such variability, it would be great to have RNA seq data on controls versus treated individuals able to regenerate versus treated individuals that are not able to regenerate to approach the mechanistic underlying successfully induced regeneration.

We agree, and are currently pursuing sequencing experiments as a next step to probe the molecular mechanisms. For the scope of this paper that focuses on establishing regeneration stimulation and its comparative presence across animals, we believe the data presented support the current conclusions.

L86: "it would more likely be intact" what do you mean here?

We have in mind an analogy with how an organ that has fallen out of use could over time become vestigial. The following clarification has been added (L89):

“There is no guarantee that regeneration can be induced, e.g., because the organ never evolved the ability to regenerate or because regeneration that has been inactivated could be rendered vestigial over time. We reasoned that if there was an ancestral mechanism to promote regeneration, it would more likely be intact in early-branching lineages with prevalent regeneration across the species.”

L89: Some medusa regenerate very well complex structures and organs, see old literature and recent paper Sinigaglia et al. 2020 eLife.

We added qualification to the sentence and the citation to the Sinigaglia et al., as well as older work (L97):

“In contrast to the polyps’ ability to regenerate, regeneration in ephyrae and medusae appears more restricted in some species (Abrams et al., 2015; Sinigaglia et al. 2020; Schmid and Alder 1984).”

L113 and following: It is not clear for me the kind of amputation you did. Did you remove a single arm or more? Did you perform always the same king of amputation? If you amputate one or more arms is it changing the percentage of individuals able to regenerate?

We tested various kinds of amputations, as described in Abrams et al., 2015 (doi.org/10.1073/pnas.1502497112). We clarify this in the revised text (L121):

“Intriguingly, in the course of our previous study (Abrams et al., 2015), we observed in a few symmetrizing ephyrae, a small bud appearing at the amputation site..…”

No regeneration was observed, even from one-arm amputation (doi.org/10.1073/pnas.1502497112).

L114: Why did you do your experiments in the original habitat?

We were wondering if there were differences between the lab vs natural environment that influence the way the animals respond to injury.

L116: There is not figure 1e in the figure 1.

Thank you, this has been changed to Figure 1c (L125).

L125: Why did you amputate half of the ephyrae and not a single arm. Are you still in the context of appendage regeneration when removing half of the body?

We tested various amputation schemes, and chose the amputation scheme that was the fastest to do and facilitated consistency, enabling testing thousands of ephyrae. Removing 1 arm takes quite a bit more time as one has to cut specifically across the arm base while avoiding injuring the surrounding arms. Removing 2 arms is less hard but still requires awkward positioning of the knife. Removing 4 arms again takes more time because one has to cut through the large protruding manubrium, and this way of amputation also affects the animal’s feeding ability. This rationale is now added in Methods (L614-6120).

With the amputation scheme chosen, arm regeneration is the most obvious outcome. However, the reviewer is correct that the body is also partially regenerated. We include this point in the discussion (L502-505):

“Notably in *Aurelia*, the amputation bisected through the body, and more than appendage was in fact regenerated, e.g., circular muscle in the body is regrown (Figure 2e). Thus, nutrient supplementation may have regenerative effect in body parts beyond appendage..… ”

L121 and so on: How did you chose your "various molecular and physical factors"?L163: no effects observed by using modulators of developmental pathways. How was it done precisely?

We include in the revised legend of Figure 2—figure sup 1 a more detailed protocol of the screening experiments. Briefly, essentially the same experimental designs were used as those described for the main experiments.

We include in the revision a source file “Aurelia screen.xls” that lists in more details the screen parameters: (i) The rationale for testing the factors; (ii) The consensus mechanism of actions of the drugs, and references; (iii) The doses or range of parameters tested, and references; (iv) The number of the ephyrae tested.

L167: Is it unexpected to have water current necessity for an ephyra?

No and yes. No, if one thinks that ephyrae, being planktonic, should naturally experience water current. Yes, because from other experiments in the lab, we knew that ephyrae are perfectly fine, can recover from injury through symmetrization, and can even develop into medusae in stagnant water. Therefore, the requirement for current may be more subtle than simply for ensuring general health. We added a sentence to highlight this point in the revised manuscript (now L175):

“We first identified a necessary condition: water current. […] Behaviorally, the presence of current…”

L183: Any dose effect for addition of insulin?

We did preliminary testing with 500 nM and 3 μM, and did not see dramatic differences. 500 nM was chosen for further quantitation because we saw solubility problems at 3 μM.

Figure 3: legends are missing to help the reader to better understand the figure.

A legend has been added, thank you.

L252: what about adding leucine to HF?

We did not do this because our question was whether leucine could recapitulate the effects of high nutrients, however we are currently investigating these interactions.

L279: why not testing leucin and insulin alone before adding both in the *Drosophila* food?

We wanted the strongest stimulation to maximize the chance of seeing induction.

DiscussionL451: I am not sure to understand how the fact that regenerations at the metazoan scale rely on either stem cells or differentiated cells is supporting the idea of commonality and ancestry and upstream shared regulation.

The reliance of regeneration across species on stem cells or cells de-differentiating suggests a plausible point of commonality. Regulations of stem-ness and differentiation are tightly linked to regulation of proliferation, which is highly conserved across metazoan cells. We revised the sentences to hopefully better explain this idea (L510-515):

“… In this view of regeneration, upstream from tissue-specific morphogenesis is a conserved regulation of cell growth and proliferation. […] We propose that in animals that poorly regenerate, high nutrient input turns on growth and anabolic states that promote tissue rebuilding upon injury.”

L456: Is there any previous study showing the importance of nutrient for regeneration?

There are very few, to our knowledge, and we have cited the most relevant ones in the mansucript: The ctenophore study (Bading et al. 2017, line 549) and, as suggested by the Reviewers, the recent *Xenopus* study (Williams et al. 2021, line 555).

L457: Do we have insight about the importance of environment for regeneration in some model species?

We are not aware of studies showing that the model species do not regenerate in some environments.

L465 to 470: I am not sure to follow this part of the discussion.

What we have in mind is to contrast the variable regenerative response with how wild-type traits tend to show robustness (e.g, Waddington, 1942 https://doi.org/10.1038/150563a0; Andreas Wagner, 2007 ISBN 9780691134048). Wild-type phenotypes can of course vary too, but the variation that we observed does not resemble stable polymorphisms. Rather, it is reminiscent to variation exhibited by a mutant phenotype, which by virtue of moving out of the wild-type optimum, is sensitive to physiological parameters. We revised the sentences to better clarify this point (L528-532):

“… The variability stands in stark contrast to the robust regeneration in e.g., axolotl, planaria, or hydra. […] Thus, just like mutant phenotypes show varying penetrance and expressivity, the variable regenerative response speaks to us as a fundamental consequence of activating a latent biological module. …”

L472: Importance of food availability and so nutrient dependence have been described for ctenophore a highly regenerating model. see Bading et al. 2017. I am notIs there any previous data, such as RNA-seq data of regeneration that may point out on a role of energy/nutrient etc on regeneration?

We have cited Bading et al., 2017 (L585). No studies that we know of has implicated nutrient/energy specifically. However, the roles of metabolic parameters are increasingly being implicated in some systems, e.g., in tissue repair (doi: 10.1016/j.cell.2013.09.059), in heart regeneration (doi: 10.7554/*eLife*.50163) and more recently in zebrafish tail regeneration (doi: 10.1101/2020.03.03.975318). We find these studies tantalizing. However since we do not have specific molecular mechanisms and because these studies are in different organs, we refrain for now from citing them and making generalization. We are open to suggestion if the Reviewer thinks it would benefit the mansucript to cite these studies.

[Editors' note: further revisions were suggested prior to acceptance, as described below.]

We appreciate that you increased the sample size for experiments involving *Drosophila*. This has greatly strengthened your study. The revisions below would help to further increase the clarity of the manuscript.Essential revisions:1) Line 47: "Although the notion of convergences still cannot be fully excluded (e.g., Lai and Aboobaker, 2018), these findings begin to build the case that, rather than many instances of convergence, the ability to regenerate is ancestral (Sanchez-Alvarado, 2000; Bely and Nyberg, 2010)".We would suggest to replace it with "Although the hypothesis of convergent evolution cannot be fully excluded (e.g., Lai and Aboobaker, 2018), these findings begin to build the case that the ability to regenerate may be ancestral (Sanchez-Alvarado, 2000; Bely and Nyberg, 2010).

We have revised the sentence as suggested, thank you, this reads better.

2) Line 89-91: The following sentence is difficult to understand: "There is no guarantee that regeneration can be induced, e.g., because the organ never evolved the ability to regenerate or because regeneration that has been inactivated could be rendered vestigial over time." We would suggest to remove it and change in the next sentence: "it would more likely be intact" by "it would likely remain intact".

We agree. We removed the sentence. The first sentence of the paragraph now reads (now line 90):

“We reasoned that if there was an ancestral mechanism to promote regeneration, it would likely remain intact in early-branching lineages ….”

3) Line 122: replace "a small bud would appear at the amputation site" by "a small bud at the amputation site".

This has been revised as suggested (now line 121).

4) Line 324-325: "the tips stained positively with DAPI instead": what is the proportion of tips that stained positively? How many tips were examined in total?

This information has been provided in the legend of Figure 4f. We now include it in the main text to increase clarity (line 325):

“…the tips stained positively with DAPI instead (14 of 16 tibias examined; Figure 4f).”

5) Line 359-360: "5 flies showed reformed tibia segments". Please indicate again the total number of flies that were examined.

The total number of flies examined has been included again in the sentence (line 359):

“… 5 flies showed reformed tibia segments (N=150; Figure 5h), … “

6) Line 706: "Treated flies were supplemented with yeast." And what about the control flies?

We revised the sentence to clarify:

“Treated flies, but not control flies, were sometime continued on yeast supplementation.”

7) Line 991: " Experiments were performed at 68°F". Please use °C.

We have changed the temperature to Celsius. “Experiments were performed at 20°C…”